# Semi-Empirical Objective Functions for Neural MCMC Proposal Optimization

## Abstract

Current objective functions used for training neural MCMC proposal distributions implicitly rely on architectural restrictions to yield sensible optimization results, which hampers the development of highly expressive neural MCMC proposal architectures. In this work, we introduce and demonstrate a semi-empirical procedure for determining approximate objective functions suitable for optimizing arbitrarily parameterized proposal distributions in MCMC methods. Our proposed Ab Initio objective functions consist of the weighted combination of functions following constraints on their global optima and transformation invariances that we argue should be upheld by general measures of MCMC efficiency for use in proposal optimization. Our experimental results demonstrate that Ab Initio objective functions maintain favorable performance and preferable optimization behavior compared to existing objective functions for neural MCMC optimization. We find that Ab Initio objective functions are sufficiently robust to enable the confident optimization of neural proposal distributions parameterized by deep generative networks extending beyond the regimes of traditional MCMC schemes.

## 1 Introduction

The development of efficient Markov Chain Monte Carlo (MCMC) proposal distributions is vital to enable numerical estimation and statistical inference in increasingly complicated problem domains. If we had an exact definition of the notion of MCMC efficiency, this could be accomplished by straightforward optimization over a set of proposal distributions. Past computational limitations encouraged the use of architecturally limited MCMC schemes that yield useful results for a wide range of target distributions, such as Random Walk Metropolis (RWM) (Metropolis et al., 1953), Metroplis Adjusted Langevin Diffusion (MALA) (Besag and Green, 1993), and Hamiltonian Monte Carlo (HMC) (Duane et al., 1987). Although much research has been devoted to adaptive methods to improve MCMC performance, traditional adaptive methods (Roberts and Rosenthal, 2009; Sejdinovic et al., 2014; Haario et al., 2001; Roberts and Rosenthal, 2007) focus on optimization across highly restricted proposal distribution model classes. Ideally, we would like to develop a practical means of optimizing MCMC performance over arbitrarily parameterized classes of proposal distributions.

Given their demonstrated success in parameterizing expressive distributions, deep generative models are naturally suited to the problem of MCMC proposal optimization. Recent research regarding applications of deep learning to proposal optimization has yielded new MCMC schemes (Song et al., 2017; Spanbauer et al., 2020; Hoffman et al., 2019) and extensions of existing schemes (de Freitas et al., 2001; Habib and Barber, 2018; Levy et al., 2018; Li et al., 2021). A variety of objective functions are utilized to optimize these deep learning based approaches, with functional forms generally dependent on the type of MCMC scheme being optimized. As demonstrated by the experiments of this work, the objective functions currently used for MCMC proposal optimization rely on model class restrictions imposed on the proposal architecture and are not suitable for optimizing proposal distributions more expressive than traditional MCMC schemes. In this work, we introduce and demonstrate Ab Initio objective functions for MCMC proposal optimization intended to remain compatible with any proposal distribution defined using deep generative models.

Presumably, there exists some "ground truth" objective function underlying our notion of MCMC sampling performance. However, even after the decades of research regarding MCMC methods, there appears to be no universal definition for what we mean by MCMC efficiency. Metrics like effective

sample size (ESS) (Gelman et al., 2013; Vats et al., 2019) generally coincide with our notion of sampling performance, but no such metric serves as the canonical definition of MCMC efficiency. Theoretical analysis regarding optimal acceptance rates for particular MCMC schemes (Roberts et al., 2001; Gelman et al., 1997; Roberts and Rosenthal, 1998; Neal et al., 2012; Beskos et al., 2013) considers restricted proposal schemes and targets within a continuous diffusionary limit wherein our common performance metrics converge in their definition of optimality (Roberts et al., 2001). Within this diffusionary limit, useful properties regarding MCMC optimality (e.g. the rules of thumb to seek an acceptance rate of 0.234 when using RWM and of 0.574 when using MALA) may be derived without specifically defining sampling performance. In light of the lack of an exactly specified objective function for MCMC efficiency, our Ab Initio objective functions seek to approximate the ground truth definition of sampling performance by adhering to certain reasonable first principles properties and fitting to reproduce the "mathematical observations" provided by existing theoretical analysis of optimal acceptance rates.

Our contributions are as follows:

- We describe a set of first principles properties that may be reasonably assumed of the ground truth objective function underlying our notion of MCMC efficiency.
- We illustrate the construction of an example Ab Initio objective function via the combination of simpler objective functions with coefficients determined to reproduce optimal behavior on reference problems with analytically known solutions.
- We verify the generality of the resulting Ab Initio objective function through its ability to closely reproduce analytically known optimal results for a wide range of optimization tasks beyond the reference problem used in its construction.
- Through a series of illustrative experiments, we demonstrate the advantages of Ab Initio objective functions for optimizing arbitrarily defined MCMC proposal distributions.

## 2 RELATION TO PRIOR WORK

Ab Initio techniques (Hehre, 1976; Yin and Cohen, 1982; Marx and Hutter, 2009) are used in the physical sciences to simulate systems that would otherwise be unobservable in a laboratory setting. These Ab Initio methods are founded on principled approximations of fundamental physical laws with parameters chosen to reproduce the properties of reference systems that can be observed. We take inspiration from these methods in the physical sciences in forming the methodology of this work.

The purpose of this work is to introduce a procedure for selecting objective functions for MCMC proposal optimization that are suited to the optimization of proposal model classes arbitrarily parameterized by deep generative models. Research into the applications of deep learning for MCMC proposal optimization (Song et al., 2017; Spanbauer et al., 2020; Hoffman et al., 2019; de Freitas et al., 2001; Habib and Barber, 2018; Levy et al., 2018; Li et al., 2021) has yielded a number of potential candidates for this objective function. We therefore compare our Ab Initio objective functions to these candidates on the basis of their suitability for optimization of arbitrary proposal distributions.

Pure KL-divergence based objectives (de Freitas et al., 2001; Habib and Barber, 2018; Neklyudov et al., 2018; Wang et al., 2018) have found some success, particularly when optimizing resampling style schemes. These objectives are unable to properly optimize proposals within a diffusionary limit (Titsias and Dellaportas, 2019). We therefore omit pure KL-divergence based objectives from the comparisons within this work.

Adversarial objectives have been used to optimize proposal distributions, notably with A-NICE-MC (Song et al., 2017; Spanbauer et al., 2020). We view adversarial training as approximately optimizing an existing performance measure (e.g. KL-divergence), rather than defining a fundamentally new performance measure. We therefore also omit adversarial objectives from our comparisons.

Mean squared jump distance (Pasarica and Gelman, 2010) (MSJD) remains a popular objective for optimizing proposals. Notably, L2HMC (Levy et al., 2018) is optimized using a modification of MSJD with a regularization intended to encourage mixing of the resulting Markov Chain. As shown in our experiments, both MSJD and L2HMC's modification can produce arbitrarily non-ergodic Markov Chains when optimizing proposal distributions with position dependence. Our Ab Initio objective functions avoid this undesirable optimization behavior by maintaining i.i.d. resampling

from the target as their unique global minima. Another potential advantage of the Ab Initio objective of Equation (2) over MSJD based objective functions is that it does not rely on the notion of a metric of the underlying space of the proposal distribution, and so may be more suited to optimizing proposals in settings where the space is equipped with only a measure and not a metric (e.g. MCMC settings involving discrete or graph based distributions).

Recently, Titsias and Dellaportas (2019) introduced the Generalized Speed Measure (GSM) as an objective function for MCMC proposal optimization. The GSM amounts to maximizing proposal entropy subject to the constraint of achieving a user specified acceptance rate. Although it is theoretically well motivated, the GSM relies on knowledge of optimal acceptance rates for a given optimization problem, which will generally be unknown when using very general neural architectures. Our example Ab Initio objective function of Equation (1) takes inspiration from the components of the GSM, while also ensuring the functional limitations argued for in Section 4. A key advantage of our Ab Initio objective functions over the GSM is that they do not require prior knowledge to recover near optimal MCMC behavior (e.g. our Ab Initio objective function, whose construction only involved the knowledge of the optimal acceptance rate for RWM, recovers optimized MALA proposals with acceptance rates around 0.5). An additional advantage of our Ab Initio objective functions over the GSM is that they may be used to compare the efficiencies of proposal distributions with differing acceptance rates outside of the context of parameter optimization.

## 3 MCMC Proposal Optimization

In this work, we consider the task of optimizing a proposal density $g_\theta(\vec{\mathbf{x}}'|\vec{\mathbf{x}})$ ($\vec{\mathbf{x}} \in \mathbb{R}^n$) to sample from a target density $\pi(\vec{\mathbf{x}})$ for a MCMC task. Proposals are accepted with rate $\alpha_{g_\theta,\pi}(\vec{\mathbf{x}}'|\vec{\mathbf{x}})$ that ensures the resulting Markov Chain converges towards $\pi(\vec{\mathbf{x}})$. For this work, we restrict ourselves to Metropolis-Hastings type schemes, wherein $\alpha_{g_\theta,\pi}(\vec{\mathbf{x}}'|\vec{\mathbf{x}}) = \min\{1, \frac{\pi(\vec{\mathbf{x}}')g_\theta(\vec{\mathbf{x}}|\vec{\mathbf{x}}')}{\pi(\vec{\mathbf{x}})g_\theta(\vec{\mathbf{x}}'|\vec{\mathbf{x}})}\}$. For optimization, we utilize some measure of sampling performance to define an objective function $\mathcal{L}[g;\pi]$ that imposes an ordering over proposal distributions by defining $g_1 < g_2$ exactly when $\mathcal{L}[g_1;\pi] > \mathcal{L}[g_2;\pi]$. We do not seek to provide an argument for the application of deep learning methods to MCMC proposal optimization, so we will simply assume that all objective functions of interest are well-behaved for optimization via deep learning techniques (i.e. they are continuous and almost surely differentiable). We will also assume that all proposal and target densities considered are positive and non-singular.

Let $G$ be the set of allowed proposals to consider during optimization and let $\mathcal{D}$ be the group of almost sure diffeomorphisms over the space of our data. To perform optimization, we first select some $T \in \mathcal{D}$ that provides us with the coordinate system we will use for optimization. Defining:

$$T \circ f(\vec{\mathbf{z}}) = f(T^{-1}(\vec{\mathbf{z}}))|\frac{\partial T^{-1}(\vec{\mathbf{z}})}{\partial \vec{\mathbf{z}}}|$$

We finally optimize to find $g_{opt} = \underset{g \in G}{\mathrm{argmin}}\ \mathcal{L}[T \circ g; T \circ \pi]$.

The focus of this work is to illustrate the construction of objective functions such that, when the model class $G$ is very expansive (e.g. having been parameterized by a deep generative model), the optimized proposal $g_{opt}$ aligns with our notion of an efficient MCMC proposal, as discussed below.

## 4 Properties of the Ground Truth Objective Function $\mathcal{L}^*$

For us to sensibly pursue the task of MCMC proposal optimization, we must assume the existence of some ground truth objective, $\mathcal{L}^*$, that produces our notion of sampling performance. As previously stated, we currently do not know a universal definition for $\mathcal{L}^*$. We may, however, assume that the ground truth objective satisfies a number of first principles properties:

$\mathcal{L}^*$ **is Proper**    Define an objective function to be *proper* if it attains a unique global minimum at $g(\vec{\mathbf{x}}'|\vec{\mathbf{x}}) = \pi(\vec{\mathbf{x}}')$ and $\alpha_{g,\pi}(\vec{\mathbf{x}}'|\vec{\mathbf{x}}) = 1$ for all $\vec{\mathbf{x}}', \vec{\mathbf{x}}$. The overall goal of our MCMC methodology is to approximate perfect i.i.d. sampling from the target, $\pi$. We therefore find it uncontroversial to assume that $\mathcal{L}^*$ is proper.

$\mathcal{L}^*$ **is Representation Independent**   Define an objective function, $\mathcal{L}$, to be *representation independent* over a group of almost sure diffeomorphisms $\mathcal{T}$ if, for all $T \in \mathcal{T}$, and for all proposal distributions $g_1, g_2$, $\mathcal{L}[g_1; \pi] > \mathcal{L}[g_2; \pi]$ if and only if $\mathcal{L}[T \circ g_1; T \circ \pi] > \mathcal{L}[T \circ g_2; T \circ \pi]$. Similarly, we will say $\mathcal{L}$ is *representation invariant* over $\mathcal{T}$ if $\mathcal{L}[T \circ g; T \circ \pi] = \mathcal{L}[g; \pi]$ for all $g$. If $\mathcal{L}^*$ were not representation independent over $\mathcal{D}$, then our definition of sampling performance would depend on which $T \in \mathcal{D}/\mathcal{H}$ is used when computing the optimization, where $\mathcal{H}$ is the maximal subgroup of $\mathcal{D}$ over which $\mathcal{L}^*$ remains representation independent. To fully justify an ordering, we would need to justify our selection of a particular member from $\mathcal{D}/\mathcal{H}$, which we should expect to be exceptionally burdensome. Of course, we usually have little prior justification for selecting a particular $T$ and instead often use the coordinate system in which data was originally collected, perhaps applying some rescaling and recentering. Thus, unless contradicted by future experimental or theoretical results, we should assume that $\mathcal{L}^*$ is at least representation independent.

$\mathcal{L}^*$ **Yields Established Optimal Results**   Prior theoretical analysis has established certain properties regarding optimal proposal distributions for a number of MCMC schemes under diffusionary limits. If $\mathcal{L}^*$ is to correspond to the same notion of sampling performance, it must yield the same results. Thus, we should expect that optimization of $\mathcal{L}^*$ will recover the properties established within these theoretical works when applied in the same diffusionary limits.

## 5   CONSTRUCTING AB INITIO OBJECTIVE FUNCTIONS

Without knowing the exact form of $\mathcal{L}^*$, we must resort to finding a useful approximation. Although the assumptions of being proper and representation independent limit the functional class to which $\mathcal{L}^*$ belongs, this functional class remains quite expansive. This situation is greatly simplified by restricting our consideration to representation invariant objective functions. As shown in Appendix A, the positive weighted combination of proper and representation invariant objective functions is itself a proper and representation invariant objective function.

This allows us to construct potential approximations of $\mathcal{L}^*$ by the combination of simpler objective functions as weighted by hyperparameter coefficients. These hyperparameter coefficients can then be fit so as to recover optimal properties established within existing theoretical works. We call the resulting objective function an *Ab Initio* objective function.

For this work, we adapt the GSM into an Ab Initio objective function (details are provided in Appendix B) to follow the constraints of Section 4 and consider a functional class of the form:

$$\mathcal{L}[g; \pi] = \mathbb{E}_{\vec{\mathbf{x}} \sim \pi(\vec{\mathbf{x}})}[\, D_{KL}(g(\vec{\mathbf{x}}'|\vec{\mathbf{x}})||\pi(\vec{\mathbf{x}}')) - A\, d\, \mathbb{E}_{\vec{\mathbf{x}}' \sim g(\vec{\mathbf{x}}'|\vec{\mathbf{x}})}[\log \alpha(\vec{\mathbf{x}}'|\vec{\mathbf{x}})]\, ] \tag{1}$$

Where $A$ is a hyperparameter coefficient for fitting and $d$ is the dimensionality of the target. In Appendix C, we demonstrate that this functional class is proper and representation invariant.

## 6   FITTING AND VERIFYING AB INITIO OBJECTIVE FUNCTIONS

There are many possible approaches to fitting the coefficients of an Ab Initio objective function to recover optimal results over reference problems. General and principled methodologies for this fitting are provided by procedures like stochastic Levenberg-Marquardt optimization (Bergou et al., 2018).

For simplicity, we determined $A$ by manual approximate Newton-Raphson to match the theoretical result that RWM has an optimal acceptance rate of 0.234 when targeting a multivariate gaussian of zero mean and identity covariance. As the theoretical results are set within the diffusionary limit of $d \to \infty$, we must match the theoretical result in a problem with sufficiently large dimension in practice. Here, we perform this match to theoretical reference on a problem with 1000 dimensions. We recovered the desired acceptance rate with $A = 0.18125$ and the Ab Initio objective function considered through the remainder of this work is therefore:

$$\mathcal{L}[g; \pi] = \mathbb{E}_{\vec{\mathbf{x}} \sim \pi(\vec{\mathbf{x}})}[\, D_{KL}(g(\vec{\mathbf{x}}'|\vec{\mathbf{x}})||\pi(\vec{\mathbf{x}}')) - 0.18125\, d\, \mathbb{E}_{\vec{\mathbf{x}}' \sim g(\vec{\mathbf{x}}'|\vec{\mathbf{x}})}[\log \alpha(\vec{\mathbf{x}}'|\vec{\mathbf{x}})]\, ] \tag{2}$$

By fitting the parameters of an Ab Initio objective function, we hope to approximate a general notion of MCMC efficiency beyond the particularities of the reference problem(s) used in its construction. To verify this robustness, we utilized the Ab Initio function listed in Equation (2) to perform MCMC

Table 1: Verification results from applying Ab Initio objective function of Equation (2) to multiple MCMC optimization tasks with analytically known optimal properties. The reference problem used for coefficient fitting is shown in bold. Value means are reported to at most the first significant digit of standard error (reported in parentheses).

| | | | $d = 100$ | | $d = 1000$ | | $d = 10000$ | | |
| | | | Ab Initio Eq. (2) | Analytic Simulated | Ab Initio Eq. (2) | Analytic Simulated | Ab Initio Eq. (2) | Analytic Simulated | Analytic Exact |
|---|---|---|---|---|---|---|---|---|---|
| Reference Problem | | | | | | | | | |
| Gaussian | RWM | Acc. Rate | 0.246(2) | 0.236(1) | **0.233(5)** | **0.234(1)** | 0.228(6) | 0.235(2) | 0.234 [1] |
| | | MSJD | 1.32(2) | 1.32(1) | **1.32(1)** | **1.324(4)** | 1.33(1) | 1.33(1) | |
| Scheme Robustness | | | | | | | | | |
| Gaussian | MALA | Acc. Rate | 0.546(5) | 0.572(2) | 0.503(5) | 0.572(2) | 0.495(9) | 0.573(3) | 0.574 [2] |
| | | MSJD | 38.6(2) | 38.4(2) | 166(1) | 167(1) | 738(6) | 748(3) | |
| Target Robustness | | | | | | | | | |
| Uniform | RWM | Acc. Rate | 0.146(9) | 0.135(2) | 0.146(1) | 0.134(2) | 0.144(9) | 0.137(2) | 0.135 [3] |
| | | MSJD | 8.2(1)e-3 | 8.2(1)e-3 | 8.5(1)e-4 | 8.4(1)e-4 | 8.5(2)e-5 | 8.6(1)e-5 | |
| Laplace | RWM | Acc. Rate | 0.231(4) | 0.234(3) | 0.226(4) | 0.234(2) | 0.229(2) | 0.235(2) | 0.234 [1] |
| | | MSJD | 1.47(2) | 1.48(2) | 1.37(1) | 1.37(1) | 1.33(2) | 1.34(1) | |
| Cauchy | RWM | Acc. Rate | 0.237(3) | 0.235(2) | 0.236(5) | 0.234(1) | 0.233(3) | 0.233(2) | 0.234 [1] |
| | | MSJD | 2.72(3) | 2.72(3) | 2.67(3) | 2.64(1) | 2.67(3) | 2.61(2) | |
| | | | | | Dimensional Robustness | | | | |

optimization in a number of tasks where the optimal acceptance rate is analytically known, with varying target distribution (independent gaussian, laplace, cauchy, and uniform), dimensionality (100-10,000 dimensions), and MCMC scheme (RWM with gaussian proposals and MALA). These tasks are all within the diffusionary limit and involve a severely limited model class for proposal distributions (optimizing only proposal step size). Under these limitations, acceptance rate adequately summarizes differences between optimized model parameters and MSJD adequately summarizes the efficiency of the resulting Markov Chains. These measures are estimated for each optimized model based on 25,000 proposals from starting points independently sampled from the target distribution. To gather statistics regarding the mean and standard error of reported variables, each verification optimization is replicated a total 5 times. These results are listed in Table 1. Additional experimental details and results are provided in Appendix D.

We find that the Ab Initio objective of Equation (2) exhibits good agreement with known analytical results beyond the reference problem used in its construction. The best agreement, both in acceptance rate and resulting MSJD, is obtained in the RWM optimization tasks with continuous target densities. Even though the objective was tuned to replicate optimal behavior when optimizing RWM for a 1000 dimensional Gaussian target, the Ab Initio objective is also able to reproduce optimal behavior when optimizing a MALA proposal against a continuous target (recovering an acceptance rate around 0.5) and when optimizing RWM proposal against a discontinuous target (recovering an acceptance rate around 0.145). With one exception (the 10000 dimensional MALA optmization experiment, where the difference in MSJD is less than 2%), the difference in MSJD perfomance between the Ab Initio and analytically optimized proposals is statistically insignificant.

These results demonstrate that the Ab Initio objective remains a useful approximation of our notion of MCMC efficiency across a wide range of optization problems within the diffusionary limit for which we have analytically known solutions. At the same time, defining the Ab Initio objective to be proper ensures, in principle, valid optimization behaviour in the opposite asymptotic limit of model complexity wherein i.i.d. resampling from the target lies within the proposal distribution's model class. The Ab Initio objective is a sufficiently robust approximation to enable confident optimization of highly expressive neural MCMC proposal distributions.

---

[1] Roberts et al. (2001); Gelman et al. (1997)

[2] Roberts et al. (2001); Roberts and Rosenthal (1998)

[3] Roberts et al. (2001); Neal et al. (2012)

# 7 EXPERIMENTAL RESULTS

To demonstrate the advantages of Ab Initio objective functions, we present a series of illustrative experiments. In these experiments, we compare objective functions on the basis of their capabilities for optimizing very general parameterizations of proposal distributions that should be of interest for MCMC proposal optimization. Throughout these experiments, we use effective sample size (ESS) as a measure of sampling performance. We consider one-dimensional ESS relating to uncertainty estimating first and second moments of the target distribution's marginals (further explained in Appendix L), reporting the minimum obtained across all dimensions. For comparisons, we consider MSJD optimization (MSJD Opt.) (Pasarica and Gelman, 2010), L2HMC's objective (L2HMC Obj.) (Levy et al., 2018), and the Generalized Speed Measure (Titsias and Dellaportas, 2019) targeting acceptance rates of 0.9, 0.6, and 0.3 (GSM-90, -60, and -30). As we are comparing the properties of the global optima of these objective functions, we estimate sampling from the target distribution by either direct sampling or by sampling form a long equilibrated HMC chain prepared before optimization, as opposed to the online, persistent single chain updates common used in many adaptive MCMC algorithms. Details of the compared objective functions are provided in Appendix F. A comparison of our optimization procedure with traditional online adaptive techniques is provided in Appendix M. Additional experimental details and results are provided in Appendices G, J and K.

## 7.1 OPTIMIZING MULTI-SCHEME PROPOSALS

We now introduce a normalizing flow (Tabak and Turner, 2013; Rezende and Mohamed, 2015; Papamakarios et al., 2019) based proposal distribution. With $T_\theta$ denoting a normalizing flow's parameterized transformation, the functions $\mu_{L,\theta}$, $\mu_{D,\theta}$, $\Sigma_{L,\theta}$, and $\Sigma_{D,\theta}$ specified by neural networks, and $\odot$ denoting element wise multiplication, our proposal distribution is defined via:

$$\vec{\mathbf{n}} \sim \mathcal{N}\{\vec{\mathbf{0}} \,;\, I\}$$
$$\vec{\mathbf{z}}'|\vec{\mathbf{x}}, \vec{\mathbf{n}} = \mu_{L,\theta}(T_\theta^{-1}(\vec{\mathbf{x}})) + \Sigma_{L,\theta}(T_\theta^{-1}(\vec{\mathbf{x}})) \odot \vec{\mathbf{n}} \qquad (3)$$
$$\vec{\mathbf{x}}' = \mu_{D,\theta}(\vec{\mathbf{x}}) + \Sigma_{D,\theta}(\vec{\mathbf{x}}) \odot T_\theta(\vec{\mathbf{z}}')$$

This implementation is fundamentally a conditional normalizing flow (Winkler et al., 2019) and is similar to NeuTra-HMC (Hoffman et al., 2019), which employs HMC within the latent space of a pre-fit normalizing flow to improve MCMC efficiency. This multi-scheme distribution differs from NeuTra-HMC by allowing both the flow's transformation and the latent distribution to be optimized to improve MCMC efficiency, rather than being fixed from the start. In Appendix E, we demonstrate how this parameterization is able to approximate or recover various existing MCMC schemes.

The target distribution is an equal mixture of 4 standard gaussians positioned in a cross formation with a maximal distance of 8 between component centers. We use the NICE architecture (Dinh et al., 2014) to define the normalizing flow and vary the number of coupling layers (using either 8 or 3 layers), which influences how accurately the flow is able to approximate the target distribution. After each optimization, we determine efficiency measures on the basis of 5 Markov Chains of 1000 total proposals starting from independent samples from the target distribution. Statistics regarding the mean and standard error of these measures are collected from 5 replications for each considered objective function. The results of these optimizations are provided in Table 2. Comparisons of the computational costs[4] for optimizing each objective are provided in Table 3.

Density plots of multi-scheme proposal distributions optimized using an MSJD objective and our Ab Initio objective are provided in Figure 1. Position dependence within the proposal distribution lead MSJD and L2HMC's modification to optimize towards arbitrarily non-ergodic proposal distributions, sampling only either the horizontal or vertical components of the target distribution.

This experiment demonstrates a fundamental difficulty in applying the GSM objective to multi-scheme proposals. Here, the flow's depth determines it's maximal accuracy in approximating independent resampling of the target distribution, creating a maximum acceptance rate that can be obtained while still approximating global resampling. Attaining greater acceptance rates is possible, but forces inefficient RWM-like behavior. In this case, 8 coupling layers within the flow permits a maximal resampling acceptance rate near 0.70, while 3 coupling layers lowers this maximal rate below 0.60. A

---

[4]Calculations performed using a 2080 RTX GPU and a Xeon Gold 6152 CPU.

Table 2: Comparison of sampling performance obtained optimizing the multi-scheme proposals of Equation (3) using various objective functions. Results are reported for two choices for the number of coupling layers used in the proposal's normalizing flow ($L = 8$ and $L = 3$). Value means are reported to at most the first significant digit of standard error (reported in parentheses).

| | L=8 | | | | L=3 | | | |
| --- | --- | --- | --- | --- | --- | --- | --- | --- |
| | Acc. Rate | MSJD | ESS per Proposal | | Acc. Rate | MSJD | ESS per Proposal | |
| | | | $x_i$ | $x_i^2$ | | | $x_i$ | $x_i^2$ |
| Ab Initio, Eq. (2) | 0.67(3) | 22(1) | 0.33(5) | 0.41(5) | 0.55(4) | 17(1) | 0.18(8) | 0.2(1) |
| MSJD Opt. | 0.975(6) | 66.2(9) | 1.10(3) | 6(1)e-4 | 0.978(4) | 67(1) | 1.10(3) | 7(1)e-4 |
| L2HMC Obj. | 0.974(3) | 65.7(5) | 1.12(2) | 6(1)e-4 | 0.974(5) | 66(1) | 1.11(1) | 6.1(3)e-4 |
| GSM-90 | 0.92(1) | 1.86(5) | 5.6(4)e-4 | 7(1)e-4 | 0.90(2) | 1.80(4) | 5.3(4)e-4 | 7(1)e-4 |
| GSM-60 | 0.65(6) | 21(2) | 0.3(2) | 0.3(2) | 0.60(1) | 1.41(1) | 6(1)e-4 | 1(1)e-3 |
| GSM-30 | 0.31(1) | 10.7(4) | 0.14(2) | 0.17(1) | 0.302(6) | 10.5(4) | 0.13(2) | 0.18(2) |
| I.I.D. Resample | 1.00 | 34.4(3) | 0.98(5) | 0.99(6) | 1.00 | 33.9(3) | 0.90(7) | 1.03(5) |

Table 3: Comparison of number of gradient evaluations computed per second when optimizing the multi-scheme proposals of Equation (3) using various objective functions. Value means are reported to at most the first significant digit of standard error (reported in parentheses).

| | Ab Initio, Eq. (2) | MSJD Opt. | L2HMC Obj. | GSM |
| --- | --- | --- | --- | --- |
| L=8 | 7.6(1) | 9.9(3) | 10.3(3) | 7.33(7) |
| L=3 | 10.6(1) | 12.4(6) | 12.6(4) | 9.4(4) |

$g(x', y'|x_0, y_0)$

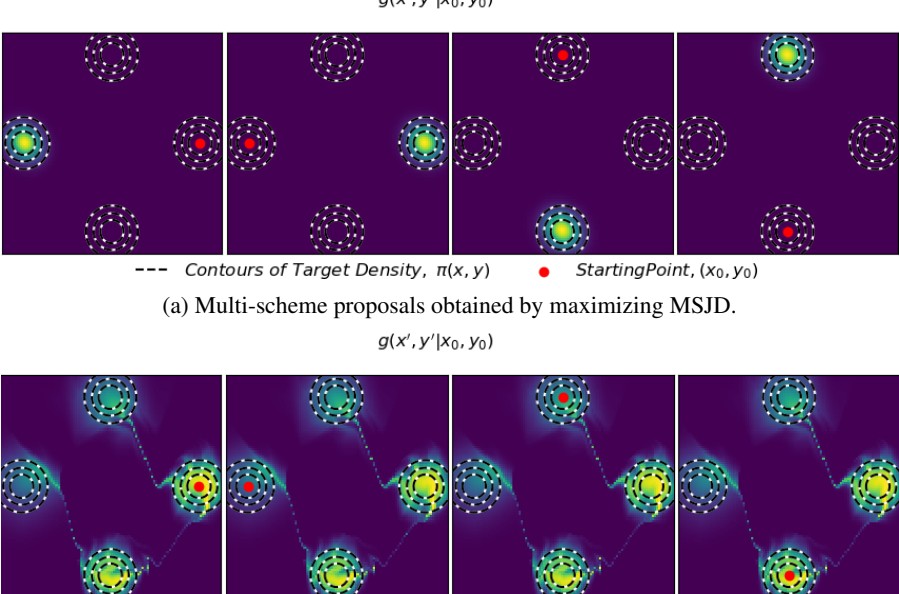

--- Contours of Target Density, $\pi(x, y)$     ● Starting Point, $(x_0, y_0)$

(a) Multi-scheme proposals obtained by maximizing MSJD.

$g(x', y'|x_0, y_0)$

--- Contours of Target Density, $\pi(x, y)$     ● Starting Point, $(x_0, y_0)$

(b) Multi-scheme proposals obtained by optimizing Ab Initio Objective.

Figure 1: Comparison of multi-scheme mixture proposals (L=8) of Equation (3) obtained by maximizing MSJD (a) and optimizing Ab Initio objective (b). Each subplot illustrates the proposal distribution's density when sampling from a given starting position (in red).

user with 8 coupling layers who uses the GSM targeting an acceptance rate of 60% to optimize their distribution would fortuitously arrive at an efficient proposal distribution. However, if they had used a

Table 4: Comparison of sampling performance obtained optimizing augmented multi-scheme proposals of Equation (3) using various objective functions targeting posterior distributions of parameters for logistic regression of UCI datasets. Value means are reported to at most the first significant digit of standard error (reported in parentheses).

| | German Credit (d=21) | | | | Heart Disease (d=14) | | | |
|---|---|---|---|---|---|---|---|---|
| | Acc. Rate | MSJD | ESS per Proposal | | Acc. Rate | MSJD | ESS per Proposal | |
| | | | $x_i$ | $x_i^2$ | | | $x_i$ | $x_i^2$ |
| Ab Initio, Eq. (2) | 0.81(1) | 0.30(1) | 0.56(3) | 0.49(7) | 0.86(2) | 1.03(4) | 0.63(6) | 0.58(7) |
| MSJD Opt. | 0.77(2) | 0.32(1) | 0.52(7) | 0.4(1) | 0.81(1) | 1.40(3) | 0.69(3) | 0.32(6) |
| L2HMC Obj. | 0.76(2) | 0.32(1) | 0.54(4) | 0.48(5) | 0.82(1) | 1.43(4) | 0.73(4) | 0.28(7) |
| GSM-90 | 0.90(1) | 0.012(1) | 0.006(1) | 0.005(1) | 0.900(3) | 0.138(3) | 0.034(2) | 0.035(1) |
| GSM-60 | 0.61(3) | 0.24(1) | 0.40(3) | 0.40(3) | 0.59(2) | 0.72(2) | 0.34(3) | 0.35(3) |
| GSM-30 | 0.29(5) | 0.13(2) | 0.13(4) | 0.13(4) | 0.29(3) | 0.42(4) | 0.13(2) | 0.14(3) |
| I.I.D. Resample | 1.00 | 0.3700(3) | 0.98(1) | 0.983(3) | 1.00 | 1.222(1) | 0.97(1) | 0.97(2) |

Table 5: Comparison of number of gradient evaluations computed per second when optimizing the multi-scheme proposals of Equation (3) using various objective functions targeting posterior distributions of parameters for logistic regression of UCI dataset. Value means are reported to at most the first significant digit of standard error (reported in parentheses).

| | Ab Initio, Eq. (2) | MSJD Opt. | L2HMC Obj. | GSM |
|---|---|---|---|---|
| German Credit | 11.5(6) | 15(1) | 14.8(9) | 10.1(3) |
| Heart Disease | 11.7(4) | 15.0(3) | 14.7(7) | 10.8(3) |

model with 3 coupling layers instead and targeted the same 60% acceptance rate, the result would be a proposal distribution far more inefficient than the actual capabilities of their proposal model class.

In both cases, the Ab Initio objective is able to optimize towards efficient approximations of i.i.d. resampling. An intuitive explanation for the results of this experiment is provided in Appendix H.

As this experiment utilizes a complicated proposal distribution, it is reasonable to question whether the problems exhibited here arise from some peculiarities of the model class of Equation (3). For this reason, Appendix I repeats a similar experiment with a simpler proposal distribution and recovers similar results. We conclude that the effects demonstrated within Table 2 may be traced back to the choice of objective function used for optimization.

## 7.2 OPTIMIZING AUGMENTED MULTI-SCHEME PROPOSALS

Augmentation with auxiliary variables (e.g. the momenta of HMC) is a common feature of MCMC schemes (Levy et al., 2018; Habib and Barber, 2018) and, in this experiment, we augment the original distributions with a number of independent auxiliary variables equal to the distribution's original dimensionality. In Appendix E.1, we explain how this augmentation enables the model class of Equation (3) approximate HMC. To demonstrate the application of Ab Initio objectives to MCMC optimization tasks of a more practical nature, we consider the optimization of the multi-scheme proposals defined in Equation (3) for MCMC sampling of regression weights in bayesian logisitc regression for various UCI datasets (Bache and Lichman, 2013). After each optimization, we determine efficiency measures on the basis of 5 Markov Chains of 20000 total proposals starting from samples drawn from a long, equilibrated Markov Chain obtained using tuned HMC. Statistics regarding the mean and standard error of these measures are collected from 5 replications for each considered objective function. The results of these optimizations are provided in Table 4. Comparisons of the computational costs[5] for optimizing each objective are provided in Table 5. Under these expanded conditions, we find the objective functions exhibiting much of the same behavior as seen in Section 7.1, in particular with the Heart Disease dataset.

---

[5]Calculations performed using a 2080 RTX GPU and a Xeon Gold 6152 CPU.

Table 6: Comparison of sampling performances and Ab Initio losses (lower is better) obtained from various MCMC schemes for the logistic regression of MNIST digits (d=785). Value means are reported to at most the first significant digit of standard error (reported in parentheses).

| Model | Obj. Func. | Acc. Rate | MSJD | ESS Per Proposal | | Ab Initio Eq. (2) | Samples Per Sec. |
|---|---|---|---|---|---|---|---|
| | | | | $x$ | $x^2$ | | |
| Precond. MALA | Eq. (2) | 0.501(7) | 129(2) | 4(1)e-6 | 5(1)e-6 | 1.644(2)e3 | 115(4) |
| Precond. MALA | MSJD | 0.543(3) | 241(1) | 1(1)e-10 | 2(2)e-10 | 2.59(3)e3 | 113(4) |
| Precond. RWM | Eq. (2) | 0.239(4) | 1.36(2) | 7(2)e-7 | 1.1(3)e-6 | 2.653(1)e3 | 205(4) |
| Multi-Scheme, Eq. (3) | Eq. (2) | 0.24(4) | 1.2(1) | 5(2)e-7 | 6(3)e-7 | 2.83(7)e3 | 47.5(4) |
| Resampling Norm. Flow | NLL | 1e-3-1e-6 | <4.73 | – | – | 5.2(3)e5 | 68.2(2) |

## 7.3 APPLICATION TO MCMC SCHEME SELECTION

As a general measure of MCMC efficiency, the Ab Initio objective funtion of Equation (2) may be used to evaluate the performance of any MCMC proposal distributions. In this experiment, we optimize a variety of different MCMC schemes for MCMC sampling of regression weights in bayesian logisitc regression between two MNIST (LeCun et al., 1998) digits. For each evaluated scheme, after optimization using the respective objective function, we determine efficiency measures on the basis of 100 Markov Chains of 1000 total proposals starting from samples drawn from a long, equilibrated Markov Chain obtained using tuned HMC. Statistics regarding the mean and standard error of these measures are collected from 5 replications for each considered objective function. The results of these optimizations are provided in Table 6. The resampling normalizing flow (an unconditional normalizing flow optimized to directly approximate the target distribution) did not produce a single accepted proposal within these evaluation Markov Chains, hence we are unable to estimate ESS and our estimates of acceptance rates and MSJD for this proposal distribution should be viewed as highly uncertain. Although preconditioned MALA proposals have a restricted model class, optimization with MSJD yields exceptionally low minimum ESS results as sampling across certain subspaces of the distribution are sacrificed in order to maximize overall MSJD. This problem is related to the *representation dependence* of MSJD, which is alleviated by the *representation independence* of Ab Initio objective functions as discussed in Section 4. Overall, we find the example Ab Initio objective function of Equation (2) to be a robust measure of MCMC efficiency that may be used to evaluate and compare the efficiency of multiple MCMC schemes in addition to its utility as an objective function for optimizing the individual parameters of highly expressive MCMC schemes.

## 8 CONCLUSION AND FUTURE WORK

In this work, we have shown how to construct Ab Initio objective functions that are suited for the optimization of highly expressive proposal distributions. We find that Ab Initio objectives are suitably robust to enable the optimization of neural MCMC proposal distributions. Our experimental results show that Ab Initio objective functions can maintain favorable performance and preferable optimization behavior compared to existing objective functions (Levy et al., 2018; Titsias and Dellaportas, 2019; Pasarica and Gelman, 2010). By design, Ab Initio objective functions are approximations of our notion of MCMC efficiency and we do not presume that the particular example Ab Initio objective function of Equation (2) will be absolutely universal. However, should future experimental or theoretical analysis demonstrate some sub-optimal properties of Equation (2), our proposed Ab Initio procedure allows for further improvements by considering alternative component functions and coefficient fitting procedures, which we plan to explore in future work.

The experimental methodology this work is intended to isolate fundamental effects of the choice of objective function on MCMC optimization. How to best train proposals distributions in an online adaptive setting, how to attain computationally efficient samples, and how to handle scaling into higher dimensions remain important practical considerations for MCMC optimization. We argue that these considerations are primarily influenced by proposal architecture and training procedure selection. We will therefore investigate these questions in a future work focusing on comparisons among specific proposal architectures and training procedures.

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

## A  COMBINING PROPER AND REPRESENTATION INVARIANT FUNCTIONS

As part of our Ab Initio approach, we rely on the following proposition:

**Proposition:**  Every positive weighted combination of proper and representation invariant objective functions is itself a proper and representation invariant objective function.

To prove this, assume that we have two proper and representation invariant (over a group of almost sure diffeomorphisms $\mathcal{D}$) objective functions $f$ and $h$ with target $\pi$ and allowed set of proposals $\mathcal{G}$. Let us select arbitrary constants $c_1, c_2 > 0$ and consider the properties of the objective function $c_1 f + c_2 h$. From the invariance of $f$ and $h$ we may immediately conclude:

$$
\begin{aligned}
(c_1 f + c_2 h)[T \circ g; T \circ \pi] &= c_1 f[T \circ g; T \circ \pi] + c_2 h[T \circ g; T \circ \pi] \\
&= c_1 f[g; \pi] + c_2 h[g; \pi] \\
&= (c_1 f + c_2 h)[g; \pi]
\end{aligned}
$$

Hence $c_1 f + c_2 h$ is itself representation invariant. Finally, for all $g(x'|x) \neq \pi(x')$:

$$
\begin{aligned}
(c_1 f + c_2 h)[\pi; \pi] &= c_1 f[\pi; \pi] + c_2 h[\pi; \pi] \\
&< c_1 f[g; \pi] + c_2 h[g; \pi] = (c_1 f + c_2 h)[g; \pi]
\end{aligned}
$$

Hence $c_1 f + c_2 h$ is also proper. This concludes the proof of the above proposition.

This offers a straightforward approach to approximating our ground truth objective $\mathcal{L}^*$. Although we do not have a guarantee that $\mathcal{L}^*$ lies within the class of representation invariant objective functions (we have only assumed representation independence), we also have too few analytical results to empirically conclude that $\mathcal{L}^*$ is not representation invariant. As fitting the coefficients of an Ab Initio objective function to match analytical results is more easily accomplished than deriving a particular objective function that happens to match analytical results, we argue that our Ab Initio approach is a practical method for obtaining approximate objective functions suitable for the optimization of arbitrary MCMC proposal distributions.

Next, we slightly extend this result in order to justify the functions composing Equation 1 of the main paper. Below, we start with a definition.

**Definition:**  An objective function is said to be *nearly proper* if it attains a (not necessarily unique) global minimum at $g(\vec{x}'|\vec{x}) = \pi(\vec{x}')$ and $\alpha_{g,\pi}(\vec{x}'|\vec{x}) = 1$ for all $\vec{x}', \vec{x}$.

We may now consider a minor generalization of the previous proposition:

**Proposition:**  Every positive weighted combination of at least one proper and representation invariant objective function with any number of nearly proper and representation independent functions is itself a proper and representation invariant objective function.

The proof that the resulting combination is representation invariant follows from the representation invariance of the component functions, exactly as in the above. To show that the combination is proper, assume that we have representation invariant (over a group of almost sure diffeomorphisms $\mathcal{D}$) objective functions $f$ and $h$ with target $\pi$ and allowed set of proposals $\mathcal{G}$. Let $f$ be proper and let $h$ be nearly proper. Let us select arbitrary constants $c_1, c_2 > 0$ and consider the properties of the objective function $c_1 f + c_2 h$. For all $g(x'|x) \neq \pi(x')$:

$$(c_1 f + c_2 h)[\pi; \pi] = c_1 f[\pi; \pi] + c_2 h[\pi; \pi]$$
$$\leq c_1 f[\pi; \pi] + c_2 h[g; \pi]$$
$$< c_1 f[g; \pi] + c_2 h[g; \pi] = (c_1 f + c_2 h)[g; \pi]$$

Hence $c_1 f + c_2 h$ is also proper.

## B    DERIVATION OF OUR EXAMPLE AB INITIO OBJECTIVE FUNCTION

We begin with the definition of the (maximized) lower bound of the GSM, where $B$ is a hyperparameter to be adapted during optimization to match the user specified acceptance rate:

$$\mathcal{L}[g; \pi] = \mathbb{E}_{\vec{\mathbf{x}} \sim \pi(\vec{\mathbf{x}})}[- \mathbb{E}_{\vec{\mathbf{x}}' \sim g(\vec{\mathbf{x}}'|\vec{\mathbf{x}})}[\log g(\vec{\mathbf{x}}'|\vec{\mathbf{x}})] + B \, \mathbb{E}_{\vec{\mathbf{x}}' \sim g(\vec{\mathbf{x}}'|\vec{\mathbf{x}})}[\log \alpha(\vec{\mathbf{x}}'|\vec{\mathbf{x}})]\,]$$

Given the success of the GSM for optimizing proposal distributions when the user knows the optimal acceptance rates for the particular problem, these components are natural choices for inclusion into an Ab Initio objective function. The addition of an expected cross entropy term, $\mathbb{E}_{\vec{\mathbf{x}} \sim \pi(\vec{\mathbf{x}})}[\,\mathbb{E}_{\vec{\mathbf{x}}' \sim g(\vec{\mathbf{x}}'|\vec{\mathbf{x}})}[\log \pi(\vec{\mathbf{x}}'|\vec{\mathbf{x}})]$, and factoring out an overall negative sign yields an objective function to be minimized that follows the functional constraints listed in Section 4:

$$\mathcal{L}[g; \pi] = \mathbb{E}_{\vec{\mathbf{x}} \sim \pi(\vec{\mathbf{x}})}[\, D_{KL}(g(\vec{\mathbf{x}}'|\vec{\mathbf{x}})||\pi(\vec{\mathbf{x}}')) - B \, \mathbb{E}_{\vec{\mathbf{x}}' \sim g(\vec{\mathbf{x}}'|\vec{\mathbf{x}})}[\log \alpha(\vec{\mathbf{x}}'|\vec{\mathbf{x}})]\,]$$

However, we encounter an additional complication. If $\mathbb{E}_{\vec{\mathbf{x}} \sim \pi(\vec{\mathbf{x}})}[\, D_{KL}(g(\vec{\mathbf{x}}'|\vec{\mathbf{x}})||\pi(\vec{\mathbf{x}}'))]$ and $\mathbb{E}_{\vec{\mathbf{x}} \sim \pi(\vec{\mathbf{x}})}[\mathbb{E}_{\vec{\mathbf{x}}' \sim g(\vec{\mathbf{x}}'|\vec{\mathbf{x}})}[\log \alpha(\vec{\mathbf{x}}'|\vec{\mathbf{x}})]\,]$ have differing dependencies on dimensionality as $d \to \infty$, then optimization with this Ab Initio objective function will incorrectly optimize towards acceptance rates of either 0 or 1 in the diffusionary limit for our chosen reference problem, as either of the two terms will completely dominate the other. To prevent this, we must introduce a dimension dependent function $A(d)$ that balances the asymptotic behavior of the two components. This yields an Ab Initio function of the form:

$$\mathcal{L}[g; \pi] = \mathbb{E}_{\vec{\mathbf{x}} \sim \pi(\vec{\mathbf{x}})}[\, D_{KL}(g(\vec{\mathbf{x}}'|\vec{\mathbf{x}})||\pi(\vec{\mathbf{x}}')) - C(d) \, \mathbb{E}_{\vec{\mathbf{x}}' \sim g(\vec{\mathbf{x}}'|\vec{\mathbf{x}})}[\log \alpha(\vec{\mathbf{x}}'|\vec{\mathbf{x}})]\,]$$

Future research may use asymptotic analysis to derive functional forms for $C(d)$ that are best suited for exceedingly high dimensions. In this work, we followed a procedure of manual trial and error and found that $C(d) = A\,d$ yields useful results at least up to $10,000$. This completes our adaptation of the GSM into the Ab Initio objective function listed in Equation (1):

$$\mathcal{L}[g; \pi] = \mathbb{E}_{\vec{\mathbf{x}} \sim \pi(\vec{\mathbf{x}})}[\, D_{KL}(g(\vec{\mathbf{x}}'|\vec{\mathbf{x}})||\pi(\vec{\mathbf{x}}')) - A\,d\, \mathbb{E}_{\vec{\mathbf{x}}' \sim g(\vec{\mathbf{x}}'|\vec{\mathbf{x}})}[\log \alpha(\vec{\mathbf{x}}'|\vec{\mathbf{x}})]\,]$$

As the Ab Initio objective function of Equations (1) and (2) functionally differs from the GSM by the addition of a cross-entropy term and by a dimensional scaling of the multiplicative coefficient, we should expect that their computational costs and complexities to be comparable. This is supported by our results in Tables 3, 5, 12, and 14-18. While the GSM is often slightly more computationally costly, we believe this is more due to our particular implementation of the online update of its hyperparameter for tuning to match the targeted acceptance rate as opposed to reflecting a fundamentally different complexity scaling of the GSM compared to the Ab Initio objective function.

## C  Properties of Our Example Ab Initio Objective Function

The functional form of our example Ab Initio objective function is given by:

$$\mathcal{L}[g;\pi] = \mathbb{E}_{\vec{\mathbf{x}}\sim\pi(\vec{\mathbf{x}})}[\, D_{KL}(g(\vec{\mathbf{x}}'|\vec{\mathbf{x}})||\pi(\vec{\mathbf{x}}')) - C(d)\,\mathbb{E}_{\vec{\mathbf{x}}'\sim g(\vec{\mathbf{x}}'|\vec{\mathbf{x}})}[\log\alpha(\vec{\mathbf{x}}'|\vec{\mathbf{x}})]\,]$$

Where $C(d)$ is a dimension dependent positive constant. We will now consider the following proposition:

**Proposition:**  The example Ab Initio objective function, $\mathcal{L}[g;\pi]$, defined above is a proper and representation independent objective function for all $C(d) \geq 0$.

Following the general restrictions we've used to define our optimization problem, let $g$ and $\pi$ be positive and smooth and let $T$ denote any almost sure diffeomorphism. Let $\mathcal{X}_T$ and $\mathcal{Z}_T$ denote the regions of the domain and range of $T$ over which $T$ is differentiable and invertible. From the properties of KL-Divergence, we know that $\mathbb{E}_{\vec{\mathbf{x}}\sim\pi(\vec{\mathbf{x}})}[\, D_{KL}(g(\vec{\mathbf{x}}'|\vec{\mathbf{x}})||\pi(\vec{\mathbf{x}}'))] \geq 0$ and is equal to 0 if and only if $g(\vec{\mathbf{x}}'|\vec{\mathbf{x}}) = \pi(\vec{\mathbf{x}}')$ for all $\vec{\mathbf{x}}, \vec{\mathbf{x}}'$. From the definition of KL-Divergence and the law of the unconscious statistician, we have:

$$\begin{aligned}
\mathbb{E}_{\vec{\mathbf{x}}\sim\pi(\vec{\mathbf{x}})}[\, D_{KL}(g(\vec{\mathbf{x}}'|\vec{\mathbf{x}})||\pi(\vec{\mathbf{x}}'))] &= \mathbb{E}_{\vec{\mathbf{x}}\sim\pi(\vec{\mathbf{x}})}\Big[\int_{\mathcal{X}_T} g(\vec{\mathbf{x}}'|\vec{\mathbf{x}})\log\frac{g(\vec{\mathbf{x}}'|\vec{\mathbf{x}})}{\pi(\vec{\mathbf{x}}')}d\vec{\mathbf{x}}'\Big] \\
&= \mathbb{E}_{\vec{\mathbf{z}}\sim T\circ\pi(\vec{\mathbf{z}})}\Big[\int_{\mathcal{X}_T} g(\vec{\mathbf{x}}'|\vec{\mathbf{z}})\log\frac{g(\vec{\mathbf{x}}'|\vec{\mathbf{z}})}{\pi(\vec{\mathbf{x}}')}d\vec{\mathbf{x}}'\Big] \\
&= \mathbb{E}_{\vec{\mathbf{z}}\sim T\circ\pi(\vec{\mathbf{z}})}\Big[\int_{\mathcal{Z}_T} T\circ g(\vec{\mathbf{z}}'|\vec{\mathbf{z}})\log\frac{T\circ g(\vec{\mathbf{z}}'|\vec{\mathbf{z}})}{T\circ\pi(\vec{\mathbf{z}}')}d\vec{\mathbf{z}}'\Big] \\
&= \mathbb{E}_{\vec{\mathbf{x}}\sim T\circ\pi(\vec{\mathbf{x}})}[\, D_{KL}(T\circ g(\vec{\mathbf{x}}'|\vec{\mathbf{x}})||T\circ\pi(\vec{\mathbf{x}}'))]
\end{aligned}$$

Hence, $\mathbb{E}_{\vec{\mathbf{x}}\sim\pi(\vec{\mathbf{x}})}[\, D_{KL}(g(\vec{\mathbf{x}}'|\vec{\mathbf{x}})||\pi(\vec{\mathbf{x}}'))]$ is a proper and representation invariant objective function.

Next, for all proposals $g$ and targets $\pi$, the Metropolis-Hastings acceptance rate is guaranteed to fall in the range $0 \leq \alpha(\vec{\mathbf{x}}'|\vec{\mathbf{x}}) \leq 1$. Additionally, if $g(\vec{\mathbf{x}}'|\vec{\mathbf{x}}) = \pi(\vec{\mathbf{x}}')$, $\alpha(\vec{\mathbf{x}}'|\vec{\mathbf{x}}) = 1$. This guarantees that $\mathbb{E}_{\vec{\mathbf{x}}\sim\pi(\vec{\mathbf{x}})}[\mathbb{E}_{\vec{\mathbf{x}}'\sim g(\vec{\mathbf{x}}'|\vec{\mathbf{x}})}[-\log\alpha(\vec{\mathbf{x}}'|\vec{\mathbf{x}})]]$ is nearly proper. And from the definition of the Metropolis-Hastings acceptance rate and the law of the unconscious statistician, we have:

$$\begin{aligned}
\mathbb{E}_{\vec{\mathbf{x}}\sim\pi(\vec{\mathbf{x}})}[\mathbb{E}_{\vec{\mathbf{x}}'\sim g(\vec{\mathbf{x}}'|\vec{\mathbf{x}})}[-\log\alpha(\vec{\mathbf{x}}'|\vec{\mathbf{x}})]] &= \mathbb{E}_{\vec{\mathbf{x}}\sim\pi(\vec{\mathbf{x}})}[\mathbb{E}_{\vec{\mathbf{x}}'\sim g(\vec{\mathbf{x}}'|\vec{\mathbf{x}})}[-\log\alpha(\vec{\mathbf{x}}'|\vec{\mathbf{x}})]] \\
&= \mathbb{E}_{\vec{\mathbf{x}}\sim\pi(\vec{\mathbf{x}})}[\mathbb{E}_{\vec{\mathbf{x}}'\sim g(\vec{\mathbf{x}}'|\vec{\mathbf{x}})}[-\log\min\{1,\frac{\pi(\vec{\mathbf{x}}')g(\vec{\mathbf{x}}|\vec{\mathbf{x}}')}{\pi(\vec{\mathbf{x}})g(\vec{\mathbf{x}}'|\vec{\mathbf{x}})}\}]] \\
&= \mathbb{E}_{\vec{\mathbf{z}}\sim T\circ\pi(\vec{\mathbf{z}})}[\mathbb{E}_{\vec{\mathbf{z}}'\sim T\circ g(\vec{\mathbf{z}}'|\vec{\mathbf{z}})}[-\log\min\{1,\frac{\pi(T^{-1}(\vec{\mathbf{z}}'))g(T^{-1}(\vec{\mathbf{z}})|\vec{\mathbf{z}}')}{\pi(T^{-1}(\vec{\mathbf{z}}))g(T^{-1}(\vec{\mathbf{z}}')|\vec{\mathbf{z}})}\}]] \\
&= \mathbb{E}_{\vec{\mathbf{z}}\sim T\circ\pi(\vec{\mathbf{z}})}[\mathbb{E}_{\vec{\mathbf{z}}'\sim T\circ g(\vec{\mathbf{z}}'|\vec{\mathbf{z}})}[-\log\min\{1,\frac{T\circ\pi(\vec{\mathbf{z}}')T\circ g(\vec{\mathbf{z}}|\vec{\mathbf{z}}')}{T\circ\pi(\vec{\mathbf{z}})T\circ g(\vec{\mathbf{z}}'|\vec{\mathbf{z}})}\}]] \\
&= \mathbb{E}_{\vec{\mathbf{x}}\sim T\circ\pi(\vec{\mathbf{x}})}[\mathbb{E}_{\vec{\mathbf{x}}'\sim T\circ g(\vec{\mathbf{x}}'|\vec{\mathbf{x}})}[-\log\alpha(\vec{\mathbf{x}}'|\vec{\mathbf{x}})]]
\end{aligned}$$

And so $\mathbb{E}_{\vec{\mathbf{x}}\sim\pi(\vec{\mathbf{x}})}[\mathbb{E}_{\vec{\mathbf{x}}'\sim g(\vec{\mathbf{x}}'|\vec{\mathbf{x}})}[-\log\alpha(\vec{\mathbf{x}}'|\vec{\mathbf{x}})]]$ is a nearly proper and coordinate invariant objective function. With $C(d) \geq 0$, our example Ab Initio objective function is therefore the positive weighted combination of a proper and representation invariant objective function with a nearly proper and representation invariant function. From the previous section, we know that this example Ab Initio objective function is itself proper and representation invariant. As representation invariance implies representation independence, this example objective function is also representation independent and thus belongs to the same functional class as our approximated "ground truth" objective.

# D    ADDITIONAL DATA FOR VERIFICATION TASKS

For verification tests, we optimize objective functions for MCMC proposals targeting isotropic gaussian, laplace, cauchy, and uniform distributions with dimensionalities between 100 and 10,000. For numerical stability, our target uniform distribution is a mixture distribution of a standard uniform distribution and a standard multivariate gaussian (with probabilities of $\frac{1}{1+e^{-100}}$ and $\frac{1}{1+e^{100}}$ respectively, to provide some non-zero likelihood beyond the support of the uniform distribution). For optimization, 20000 gradient steps are taken, with each training batch taken from a single starting point sampled i.i.d. from the target distribution from which 50 independent proposals from $g_\theta(x'|x)$ are generated to empirically estimate expected loss on the basis of 50 total samples. The gradient update steps take the form of Algorithm 1 with $M = 1$ and $N = 50$. Optimization is performed using the Adam optimizer (Kingma and Ba, 2015) with a stepsize of 3e-4 (reduced to 3e-5, 3e-6, and 3e-7 for uniform targets of dimensions 100,1000, and 10000). For optimizing GSM (Titsias and Dellaportas, 2019), $\beta$ is initially set to $d^{-1}$ and $\rho_\beta$ is set to $0.02$. Acceptance rate and MSJD are estimated for each optimized model based on 25000 proposals from starting points independently sampled from the target distribution. To gather statistics regarding the mean and standard error of reported variables, each verification optimization is replicated a total 5 times. Tables 7, 8, 9, 10, and 11 report results from optimizing all objective functions we use in the main work for comparison with our example Ab Initio objective function. In these tables, simulated analytic results are also reported, which refer to acceptance rate and MSJD as estimated in the same manner for proposals with a fixed step size determined to yield the analytically known optimal acceptance rates.

We were unable to perform gradient based optimization of MSJD and L2HMC's objective when targeting the uniform distribution, and resorted to estimating their optima by fitting curves of these objective functions with respect to acceptance rate. For MSJD, the results reported for MSJD targeting uniform distributions are therefore the result of fitting a parabola near the objective function's maxima, with uncertainties reported following the typical propagation of error. Even following this procedure, L2HMC's objective function is unable to produce non-trivial optima when targeting the uniform distribution and we therefore consider RWM optimization targeting the uniform distribution a failure case for L2HMC's objective.

Table 7: Verification results from applying MSJD maximization to the MCMC optimization tasks with analytically known optimal properties. As these verification tasks are set within the diffusionary limit, MSJD recovers known optimal results. Value means are reported to at most the first significant digit of standard error (reported in parentheses).

| | | | $d = 100$ | | $d = 1000$ | | $d = 10000$ | | |
| | | | MSJD Opt. | Analytic Simulated | MSJD Opt. | Analytic Simulated | MSJD Opt. | Analytic Simulated | Analytic Exact |
|---|---|---|---|---|---|---|---|---|---|
| Gaussian | RWM | Acc. Rate | 0.237(4) | 0.235(1) | 0.232(3) | 0.234(3) | 0.233(2) | 0.234(2) | 0.234 [6] |
| | | MSJD | 1.32(2) | 1.32(1) | 1.32(1) | 1.32(3) | 1.33(2) | 1.33(2) | |
| Gaussian | MALA | Acc. Rate | 0.538(6) | 0.574(1) | 0.559(5) | 0.573(1) | 0.56(1) | 0.574(2) | 0.574 [7] |
| | | MSJD | 38.6(1) | 38.4(1) | 167(1) | 166(1) | 749(2) | 747(5) | |
| Uniform | RWM | Acc. Rate | 0.15(2) | 0.135(5) | 0.15(2) | 0.137(4) | 0.134(2) | 0.132(3) | 0.135 [8] |
| | | MSJD | 8.2(7)e-3 | 8.2(3)e-3 | 8.4(5)e-4 | 8.6(2)e-4 | 8.4(4)e-5 | 8.2(2)e-5 | |
| Laplace | RWM | Acc. Rate | 0.21(2) | 0.234(1) | 0.23(1) | 0.233(1) | 0.24(1) | 0.234(3) | 0.234 [6] |
| | | MSJD | 1.48(1) | 1.48(1) | 1.37(2) | 1.37(0) | 1.34(1) | 1.34(2) | |
| Cauchy | RWM | Acc. Rate | 0.223(5) | 0.235(3) | 0.230(8) | 0.235(3) | 0.23(2) | 0.235(2) | 0.234 [6] |
| | | MSJD | 2.72(3) | 2.74(5) | 2.65(1) | 2.67(3) | 2.64(2) | 2.65(1) | |

---

[6] Roberts et al. (2001); Gelman et al. (1997)

[7] Roberts et al. (2001); Roberts and Rosenthal (1998)

[8] Roberts et al. (2001); Neal et al. (2012)

Table 8: Verification results from applying L2HMC's objective to the MCMC optimization tasks with analytically known optimal properties. L2HMC's objective only recovers optimal behavior for MALA proposals and cannot be used to optimize RWM proposals with a uniform target. Value means are reported to at most the first significant digit of standard error (reported in parentheses).

| | | | $d = 100$ | | $d = 1000$ | | $d = 10000$ | | |
| | | | L2HMC Obj. | Analytic Simulated | L2HMC Obj. | Analytic Simulated | L2HMC Obj. | Analytic Simulated | Analytic Exact |
|---|---|---|---|---|---|---|---|---|---|
| Gaussian | RWM | Acc. Rate | 0.587(3) | 0.233(3) | 0.580(4) | 0.235(1) | 0.56(1) | 0.233(1) | 0.234 [9] |
| | | MSJD | 0.69(1) | 1.31(1) | 0.71(1) | 1.34(1) | 0.76(4) | 1.32(1) | |
| Gaussian | MALA | Acc. Rate | 0.546(5) | 0.575(2) | 0.553(6) | 0.575(3) | 0.562(6) | 0.574(1) | 0.574 [10] |
| | | MSJD | 38.6(2) | 38.5(2) | 167(1) | 167(1) | 750(3) | 749(2) | |
| Uniform | RWM | Acc. Rate | – | – | – | – | – | – | 0.135 [11] |
| | | MSJD | – | – | – | – | – | – | |
| Laplace | RWM | Acc. Rate | 0.57(1) | 0.235(2) | 0.59(1) | 0.233(2) | 0.59(1) | 0.234(2) | 0.234 [9] |
| | | MSJD | 0.78(4) | 1.48(1) | 0.71(3) | 1.37(1) | 0.69(2) | 1.34(1) | |
| Cauchy | RWM | Acc. Rate | 0.514(4) | 0.235(2) | 0.516(5) | 0.237(1) | 0.52(1) | 0.236(1) | 0.234 [9] |
| | | MSJD | 1.77(3) | 2.71(2) | 1.77(3) | 2.69(1) | 1.74(5) | 2.66(4) | |

Table 9: Verification results from applying GSM targeting acceptance rates of 0.3 to the MCMC optimization tasks with analytically known optimal properties. Optimal behavior is approximately recovered only for RWM proposals at this acceptance rate. Value means are reported to at most the first significant digit of standard error (reported in parentheses).

| | | | $d = 100$ | | $d = 1000$ | | $d = 10000$ | | |
| | | | GSM-30 | Analytic Simulated | GSM-30 | Analytic Simulated | GSM-30 | Analytic Simulated | Analytic Exact |
|---|---|---|---|---|---|---|---|---|---|
| Gaussian | RWM | Acc. Rate | 0.298(4) | 0.235(2) | 0.295(4) | 0.234(2) | 0.299(9) | 0.234(2) | 0.234 [9] |
| | | MSJD | 1.28(1) | 1.33(2) | 1.29(1) | 1.32(1) | 1.30(1) | 1.33(1) | |
| Gaussian | MALA | Acc. Rate | 0.297(6) | 0.575(1) | 0.295(8) | 0.575(2) | 0.297(6) | 0.573(3) | 0.574 [10] |
| | | MSJD | 31.8(5) | 38.4(3) | 134(2) | 167(1) | 595(7) | 748(3) | |
| Uniform | RWM | Acc. Rate | 0.302(6) | 0.135(1) | 0.30(1) | 0.136(2) | 0.30(1) | 0.135(2) | 0.135 [11] |
| | | MSJD | 6.6(2)e-3 | 8.18(7)e-3 | 6.8(2)e-4 | 8.6(2)e-4 | 6.9(1)e-5 | 8.4(1)e-5 | |
| Laplace | RWM | Acc. Rate | 0.303(6) | 0.234(3) | 0.29(1) | 0.235(3) | 0.32(1) | 0.236(1) | 0.234 [9] |
| | | MSJD | 1.43(1) | 1.48(1) | 1.34(2) | 1.38(2) | 1.29(2) | 1.34(1) | |
| Cauchy | RWM | Acc. Rate | 0.297(2) | 0.234(3) | 0.291(3) | 0.237(1) | 0.30(1) | 0.234(2) | 0.234 [9] |
| | | MSJD | 2.62(3) | 2.71(4) | 2.58(3) | 2.69(1) | 2.57(3) | 2.65(2) | |

[9]Roberts et al. (2001); Gelman et al. (1997)

[10]Roberts et al. (2001); Roberts and Rosenthal (1998)

[11]Roberts et al. (2001); Neal et al. (2012)

Table 10: Verification results from applying GSM targeting acceptance rates of 0.6 to the MCMC optimization tasks with analytically known optimal properties. Optimal behavior is approximately recovered only for MALA proposals at this acceptance rate. Value means are reported to at most the first significant digit of standard error (reported in parentheses).

| | | | $d = 100$ | | $d = 1000$ | | $d = 10000$ | | |
|---|---|---|---|---|---|---|---|---|---|
| | | | GSM-60 | Analytic Simulated | GSM-60 | Analytic Simulated | GSM-60 | Analytic Simulated | Analytic Exact |
| Gaussian | RWM | Acc. Rate | 0.602(2) | 0.234(1) | 0.60(1) | 0.234(2) | 0.60(1) | 0.233(2) | 0.234 [12] |
| | | MSJD | 0.66(0) | 1.31(1) | 0.66(2) | 1.33(1) | 0.65(2) | 1.32(1) | |
| Gaussian | MALA | Acc. Rate | 0.607(7) | 0.575(1) | 0.60(1) | 0.575(1) | 0.596(5) | 0.576(2) | 0.574 [13] |
| | | MSJD | 38.1(2) | 38.5(2) | 166(0) | 167(0) | 743(4) | 751(4) | |
| Uniform | RWM | Acc. Rate | 0.61(1) | 0.135(2) | 0.58(1) | 0.135(1) | 0.57(4) | 0.138(1) | 0.135 [14] |
| | | MSJD | 2.4(1)e-3 | 8.20(9)e-3 | 2.7(1)e-4 | 8.52(8)e-4 | 2.8(4)e-5 | 8.64(4)e-5 | |
| Laplace | RWM | Acc. Rate | 0.60(1) | 0.234(2) | 0.61(3) | 0.233(2) | 0.63(4) | 0.237(1) | 0.234 [12] |
| | | MSJD | 0.69(3) | 1.48(1) | 0.64(6) | 1.37(2) | 0.6(1) | 1.35(1) | |
| Cauchy | RWM | Acc. Rate | 0.599(6) | 0.235(2) | 0.603(5) | 0.236(3) | 0.60(1) | 0.234(2) | 0.234 [12] |
| | | MSJD | 1.33(3) | 2.72(2) | 1.31(3) | 2.69(3) | 1.31(4) | 2.64(3) | |

Table 11: Verification results from applying GSM targeting acceptance rates of 0.9 to the MCMC optimization tasks with analytically known optimal properties. Results are significantly suboptimal. Value means are reported to at most the first significant digit of standard error (reported in parentheses).

| | | | $d = 100$ | | $d = 1000$ | | $d = 10000$ | | |
|---|---|---|---|---|---|---|---|---|---|
| | | | GSM-90 | Analytic Simulated | GSM-90 | Analytic Simulated | GSM-90 | Analytic Simulated | Analytic Exact |
| Gaussian | RWM | Acc. Rate | 0.900(3) | 0.234(2) | 0.900(2) | 0.232(2) | 0.910(6) | 0.233(2) | 0.234 [12] |
| | | MSJD | 0.06(0) | 1.31(1) | 0.06(2) | 1.32(1) | 0.05(1) | 1.32(1) | |
| Gaussian | MALA | Acc. Rate | 0.899(2) | 0.575(1) | 0.901(2) | 0.576(2) | 0.902(2) | 0.574(1) | 0.574 [13] |
| | | MSJD | 20.6(3) | 38.5(1) | 92(1) | 167(1) | 421(4) | 750(2) | |
| Uniform | RWM | Acc. Rate | 0.90(1) | 0.136(2) | 0.90(2) | 0.133(3) | 0.95(5) | 0.135(2) | 0.135 [14] |
| | | MSJD | 1.5(3)e-4 | 8.2(1)e-3 | 1.8(7)e-5 | 8.4(2)e-4 | 1(1)e-6 | 8.5(1)e-5 | |
| Laplace | RWM | Acc. Rate | 0.905(5) | 0.234(2) | 0.91(1) | 0.235(2) | 0.90(2) | 0.234(1) | 0.234 [12] |
| | | MSJD | 0.05(1) | 1.47(1) | 0.05(1) | 1.38(2) | 0.06(2) | 1.38(2) | |
| Cauchy | RWM | Acc. Rate | 0.901(4) | 0.234(1) | 0.901(2) | 0.235(2) | 0.898(5) | 0.233(2) | 0.234 [12] |
| | | MSJD | 0.11(1) | 2.73(1) | 0.11(0) | 2.65(2) | 0.12(1) | 2.65(3) | |

---

[12] Roberts et al. (2001); Gelman et al. (1997)

[13] Roberts et al. (2001); Roberts and Rosenthal (1998)

[14] Roberts et al. (2001); Neal et al. (2012)

# E    APPROXIMATING MCMC SCHEMES WITH MULTI-SCHEME PROPOSALS

As part of our experiments, we utilize a normalizing flow based multi-scheme proposal distribution, parameterized as:

$$\vec{\mathbf{n}} \sim \mathcal{N}\{\vec{\mathbf{0}} \, ; \, I\}$$
$$\vec{\mathbf{z}}'|\vec{\mathbf{x}}, \vec{\mathbf{n}} = \mu_{L,\theta}(T_\theta^{-1}(\vec{\mathbf{x}})) + \Sigma_{L,\theta}(T_\theta^{-1}(\vec{\mathbf{x}})) \odot \vec{\mathbf{n}}$$
$$\vec{\mathbf{x}}' = \mu_{D,\theta}(\vec{\mathbf{x}}) + \Sigma_{D,\theta}(\vec{\mathbf{x}}) \odot T_\theta(\vec{\mathbf{z}}')$$

Where $T_\theta$ denotes a normalizing flow's parameterized transformation, the functions $\mu_{L,\theta}$, $\mu_{D,\theta}$, $\Sigma_{L,\theta}$, and $\Sigma_{D,\theta}$ are specified by neural networks, and $\odot$ denotes element wise multiplication. This section provides examples of how this multi-scheme distribution is able to approximate many different existing MCMC schemes. Throughout, let $\pi$ denote the target distribution, $g_\theta$ denote the density of the resulting proposal distribution, and let $n$ denote the density of a standard gaussian distribution with zero mean and identity covariance.

**Independent Resampling:**    If $T_\theta \circ n \approx \pi$, $\mu_{L,\theta} = \mu_{D,\theta} = \vec{0}$, and $\Sigma_{L,\theta} = \Sigma_{D,\theta} = \vec{1}$, then:

$$g_\theta(\vec{\mathbf{x}}'|\vec{\mathbf{x}}) \approx \pi(\vec{\mathbf{x}}')$$

Depending on the accuracy of the flow's approximation, the multi-scheme distribution approximately recovers i.i.d. resampling from the target.

**Isotropic RWM:**    If $T_\theta = I$, $\mu_{L,\theta} = \vec{0}$, $\mu_{D,\theta}(\vec{\mathbf{x}}) = \vec{\mathbf{x}}$, $\Sigma_{L,\theta} = \vec{1}$, and $\Sigma_{D,\theta} = \tau$, then:

$$\vec{\mathbf{x}}'|\vec{\mathbf{x}} \sim \mathcal{N}\{\vec{\mathbf{x}}; \tau I\}$$

And the multi-scheme distribution recovers isotropic RWM with step-size $\tau$.

**Isotropic MALA:**    If $T_\theta = I$, $\mu_{L,\theta} = \vec{0}$, $\mu_{D,\theta}(\vec{\mathbf{x}}) \approx \vec{\mathbf{x}} + \tau \nabla \log \pi(\vec{\mathbf{x}})$, $\Sigma_{L,\theta} = \vec{1}$, and $\Sigma_{D,\theta} = \sqrt{2\tau}$, then:

$$\vec{\mathbf{x}}'|\vec{\mathbf{x}} \sim \mathcal{N}\{\vec{\mathbf{x}} + \tau \nabla \log \pi(\vec{\mathbf{x}}); \sqrt{2\tau} I\}$$

Depending on the accuracy of the approximation of $\mu_{D,\theta}(\vec{\mathbf{x}})$, the multi-scheme distribution recovers isotropic MALA with step-size $\tau$.

**Preconditioned Diffusions:**    If $T_\theta(\vec{\mathbf{z}}) \approx A^{-1}\vec{\mathbf{z}}$, where A is an invertible preconditioning matrix, then appropriate choices of $\mu_{L,\theta}$, $\mu_{D,\theta}$, $\Sigma_{D,\theta}$, and $\Sigma_{L,\theta}$ will approximately recover preconditioned RWM and MALA.

**Latent MALA:**    If $T_\theta \circ n \approx \pi$, $\mu_{L,\theta}(\vec{\mathbf{x}}) \approx (1+\tau)\vec{\mathbf{x}}$, $\mu_{D,\theta} = \vec{0}$, $\Sigma_{L,\theta} = \sqrt{2\tau}$, and $\Sigma_{D,\theta} = \vec{1}$, then the multi-scheme distribution will mimic the normalizing flow based proposal scheme of Hoffman et al. (2019), with MALA used within the flow's latent space rather than HMC.

## E.1    AUGMENTING VARIABLES

One method of greatly expanding the model class of proposal distribution is to augment the distribution with auxiliary variables. This is most notably accomplished with the canonical momenta in HMC. We will denote augmentation with a semicolon, $\mathbf{x}; \mathbf{a}$ indicates that $\mathbf{x}$ has been augmented with the auxiliary variables $\mathbf{a}$.

**Approximate HMC:**    If $\Sigma_{D,\theta}(\vec{\mathbf{x}}; \vec{\mathbf{p}}) \odot T_\theta(\vec{\mathbf{z}}') \approx \vec{0}$ and if $\mu_{D,\theta}(\vec{\mathbf{x}}) \approx \vec{\mathbf{x}}_{int}; \vec{\mathbf{p}}_{int}$, where $\vec{\mathbf{x}}_{int}; \vec{\mathbf{p}}_{int}$ are the results of applying any desired integrator to the system then the flow based proposal may approximate HMC by iteratively resampling $\vec{\mathbf{p}}$ from a normal distribution, generating a new proposal, and then performing a Metropolis-Hastings correction. Do note that we are not claiming that this approximation should be expected to be accurate, merely that it is mathematically possible within the flow proposal's model class. Intriguingly, the generality of the multi-scheme proposal distribution enables the construction allows for the construction of generalizations of HMC by varying the number of auxiliary variables used.

These examples are not exhaustive. Depending on implementation, these flow-based multi-scheme distributions could also parameterize useful proposal distributions that are not covered by existing MCMC schemes. An appealing aspect multi-scheme distributions is that their optimization, in theory, solves the problem of selecting which MCMC scheme to apply to a particular problem, which otherwise usually requires some intervention from the user. However, their expressiveness means that we cannot rely on guarantees that would be provided by restricting to individual MCMC schemes.

## F    DETAILS OF COMPARED OBJECTIVE FUNCTIONS

Throughout this work, we focus on comparing the properties of four objective functions for MCMC proposal optimization.

**Example Ab Initio**    From Equation (2), to be minimized:

$$\mathcal{L}[g; \pi] = \mathbb{E}_{\vec{\mathbf{x}} \sim \pi(\vec{\mathbf{x}})}[\, D_{KL}(g(\vec{\mathbf{x}}'|\vec{\mathbf{x}})||\pi(\vec{\mathbf{x}}')) - 0.18125\, d\, \mathbb{E}_{\vec{\mathbf{x}}' \sim g(\vec{\mathbf{x}}'|\vec{\mathbf{x}})}[\log \alpha(\vec{\mathbf{x}}'|\vec{\mathbf{x}})]\,]$$

**MSJD**    As used by Pasarica and Gelman (2010), to be maximized:

$$\mathcal{L}[g; \pi] = \mathbb{E}_{\vec{\mathbf{x}} \sim \pi(\vec{\mathbf{x}})}[\, \mathbb{E}_{\vec{\mathbf{x}}' \sim g(\vec{\mathbf{x}}'|\vec{\mathbf{x}})}[\, ||\vec{\mathbf{x}}' - \vec{\mathbf{x}}||^2\,]\,]$$

**L2HMC's Modification**    As proposed by Levy et al. (2018), to be minimized:

$$\mathcal{L}[g; \pi] = \mathbb{E}_{\vec{\mathbf{x}} \sim \pi(\vec{\mathbf{x}})}[\, \mathbb{E}_{\vec{\mathbf{x}}' \sim g(\vec{\mathbf{x}}'|\vec{\mathbf{x}})}[\frac{\sigma_{min}^2}{||\vec{\mathbf{x}}' - \vec{\mathbf{x}}||^2} - \frac{||\vec{\mathbf{x}}' - \vec{\mathbf{x}}||^2}{\sigma_{min}^2}\,]$$

Where $\sigma_{min}^2$ is the smallest one-dimensional variance of the target distribution. As we are primarily interested in behavior at stationarity, we omit the additional "burn-in" regularization suggested by.

**Generalized Speed Measure**    As proposed by Titsias and Dellaportas (2019), to be maximized:

$$\mathcal{L}[g; \pi] = \mathbb{E}_{\vec{\mathbf{x}} \sim \pi(\vec{\mathbf{x}})}[\mathbb{E}_{\vec{\mathbf{x}}' \sim g(\vec{\mathbf{x}}'|\vec{\mathbf{x}})}[-\beta\, \log g(\vec{\mathbf{x}}'|\vec{\mathbf{x}}) + \log \alpha(\vec{\mathbf{x}}'|\vec{\mathbf{x}})]\,]$$

Where $\beta$ is a hyper-parameter. We follow the suggested method for updating $\beta$ in an online manner. Following our discussion regarding the asymptotic behaviors of the two components with respect to dimension, we found it beneficial to initialize $\beta$ to $\frac{1}{d}$, rather than the default value of 1. After every parameter update, the $\beta$ is updated following:

$$\beta \leftarrow \beta\,(1 + \rho_\beta\,(\alpha_{target} - \alpha_{current}))$$

Where we use the suggested value of $\rho_\beta = 0.02$ throughout.

## G    FURTHER DETAILS FOR MULTI-SCHEME PROPOSAL OPTIMIZATION

Our experiments with multi-scheme proposals utilize a distribution specified as follows:

$$\vec{\mathbf{n}} \sim \mathcal{N}\{\vec{\mathbf{0}}\,;\, I\}$$
$$\vec{\mathbf{z}}'|\vec{\mathbf{x}}, \vec{\mathbf{n}} = \mu_{L,\theta}(T_\theta^{-1}(\vec{\mathbf{x}})) + \Sigma_{L,\theta}(T_\theta^{-1}(\vec{\mathbf{x}})) \odot \vec{\mathbf{n}}$$
$$\vec{\mathbf{x}}' = \mu_{D,\theta}(\vec{\mathbf{x}}) + \Sigma_{D,\theta}(\vec{\mathbf{x}}) \odot T_\theta(\vec{\mathbf{z}}')$$

The target distribution is an equal mixture of 4 standard gaussians positioned in a cross formation with a maximal distance of 8 between component centers. The additive functions, $\mu_{L,\theta}$ and $\mu_{D,\theta}$, are specified by standard ReLU networks (input dimension 2, hidden dimension 16, 4 hidden layers, output dimension 2). The multiplicative functions, $\Sigma_{L,\theta}$ and $\Sigma_{D,\theta}$, are specified by standard ReLU networks (input dimension 2, hidden dimension 16, 4 hidden layers, output dimension 2) followed by component-wise exponentiation. The normalizing flow, $T_\theta$, follows the NICE architecture (Dinh et al., 2014), with additive coupling layers specified by standard ReLU networks (input dimension 1, hidden dimension 6, 3 hidden layers, output dimension 1). For this experiment, differing numbers of coupling layers are considered ($L = 8$ and $L = 3$). For optimization, 20000 gradient steps are taken, with each training batch taken from 8 starting points sampled i.i.d. from the target distribution from each of which 100 independent proposals from $g_\theta(x'|x)$ are generated to empirically estimate

expected loss on the basis of 800 total samples. The gradient update steps take the form of Algorithm 1 with $M = 8$ and $N = 100$. Optimization is performed using the Adam optimizer (Kingma and Ba, 2015) with a stepsize of 3e-4. Optimized proposals that fall into local optima of the objective functions are discarded. For optimizing the GSM (Titsias and Dellaportas, 2019), $\beta$ is initially set to 0.5 and $\rho_\beta$ is set to 0.02 and initialization is varied to ensure the target acceptance rate is attained. After optimization, acceptance rate, MSJD, and ESS are estimated for each optimized model based on 5 independent Markov Chains of 1000 proposals with starting points independently sampled from the target distribution. To gather statistics regarding the mean and standard error of reported variables, each optimization is replicated a total 5 times. Figures 2, 3, 4, and 5 show density plots of the multi-scheme proposals obtained following optimization using L2HMC's objective Levy et al. (2018) and the GSM Titsias and Dellaportas (2019) with varying target acceptance rates. Figures 6, 7, 8, 9, 10, and 11 show density plots of the multi-scheme proposals obtained following optimization of all considered objective functions.

$$g(x', y' | x_0, y_0)$$

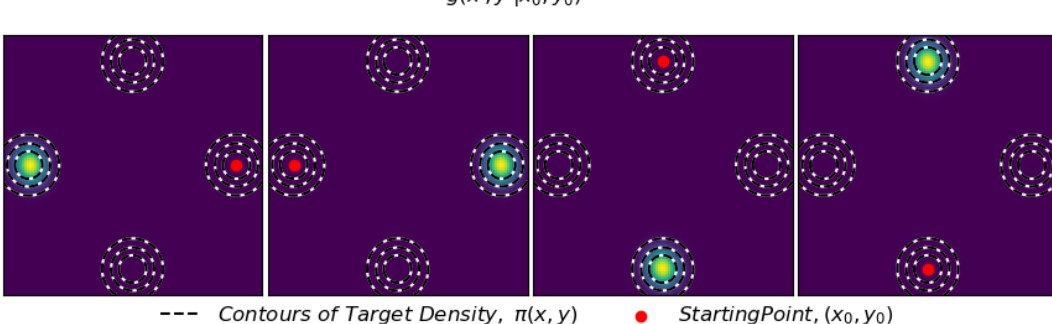

--- Contours of Target Density, $\pi(x, y)$     ● StartingPoint, $(x_0, y_0)$

Figure 2: Comparison of multi-scheme mixture proposals (L=8) of Equation 4 obtained by optimizing L2HMC's objective. Each subplot illustrates the proposal distribution's density when sampling from a given starting position (in red).

$$g(x', y' | x_0, y_0)$$

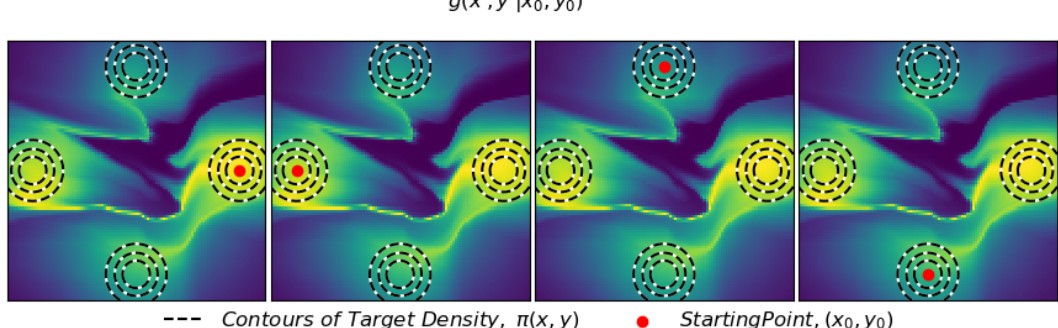

--- Contours of Target Density, $\pi(x, y)$     ● StartingPoint, $(x_0, y_0)$

Figure 3: Comparison of multi-scheme mixture proposals (L=8) of Equation 4 obtained by optimizing GSM targeting an acceptance rate of 0.3. Each subplot illustrates the proposal distribution's density when sampling from a given starting position (in red).

$$g(x', y' | x_0, y_0)$$

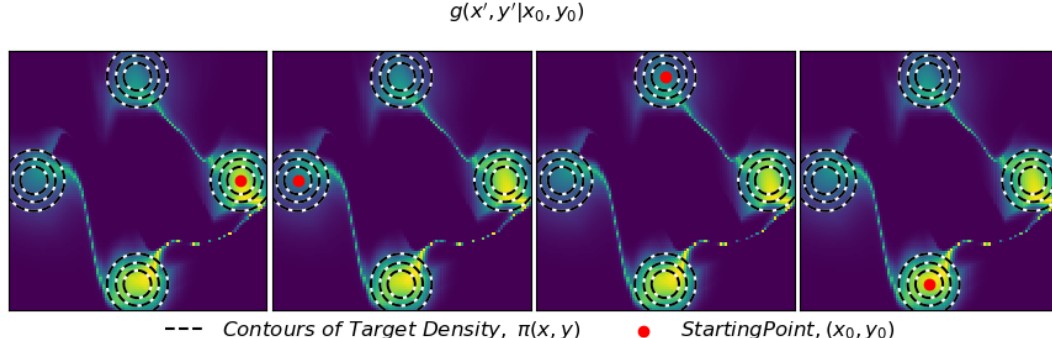

Figure 4: Comparison of multi-scheme mixture proposals (L=8) of Equation 4 obtained by optimizing GSM targeting an acceptance rate of 0.6. Each subplot illustrates the proposal distribution's density when sampling from a given starting position (in red).

$$g(x', y' | x_0, y_0)$$

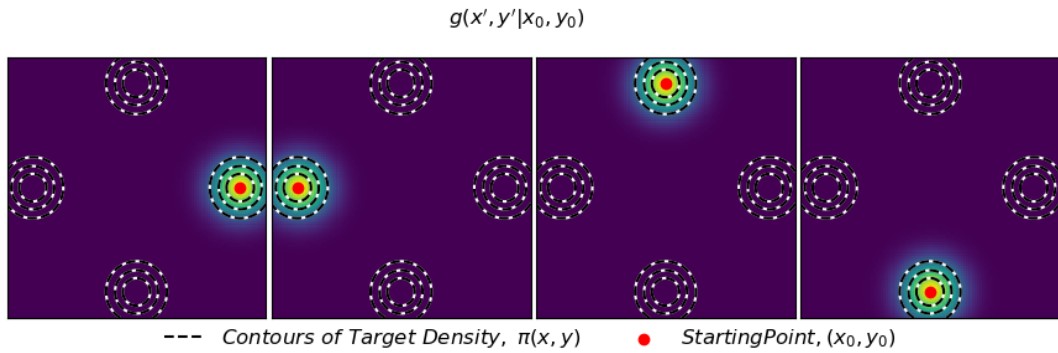

Figure 5: Comparison of multi-scheme mixture proposals (L=8) of Equation 4 obtained by optimizing GSM targeting an acceptance rate of 0.9. Each subplot illustrates the proposal distribution's density when sampling from a given starting position (in red).

$$g(x', y' | x_0, y_0)$$

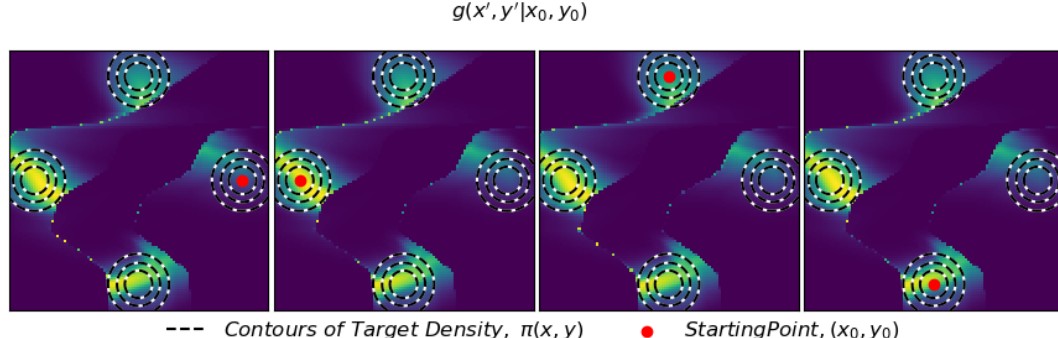

Figure 6: Comparison of multi-scheme mixture proposals (L=3) of Equation 4 obtained by optimizing our Ab Initio objective. Each subplot illustrates the proposal distribution's density when sampling from a given starting position (in red).

$$g(x', y' | x_0, y_0)$$

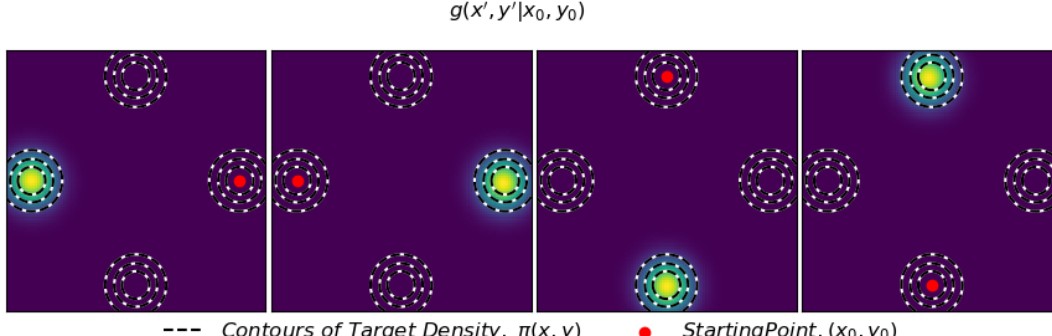

Figure 7: Comparison of multi-scheme mixture proposals (L=3) of Equation 4 obtained by maximizing MSJD. Each subplot illustrates the proposal distribution's density when sampling from a given starting position (in red).

$$g(x', y' | x_0, y_0)$$

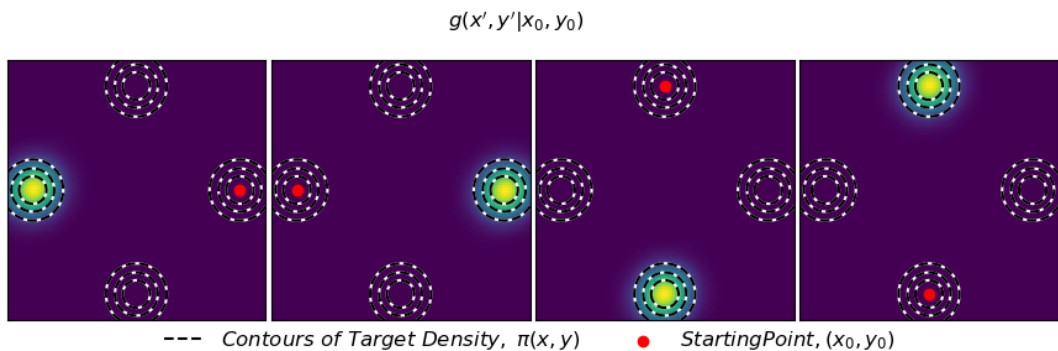

Figure 8: Comparison of multi-scheme mixture proposals (L=3) of Equation 4 obtained by optimizing L2HMC's objective. Each subplot illustrates the proposal distribution's density when sampling from a given starting position (in red).

$$g(x', y' | x_0, y_0)$$

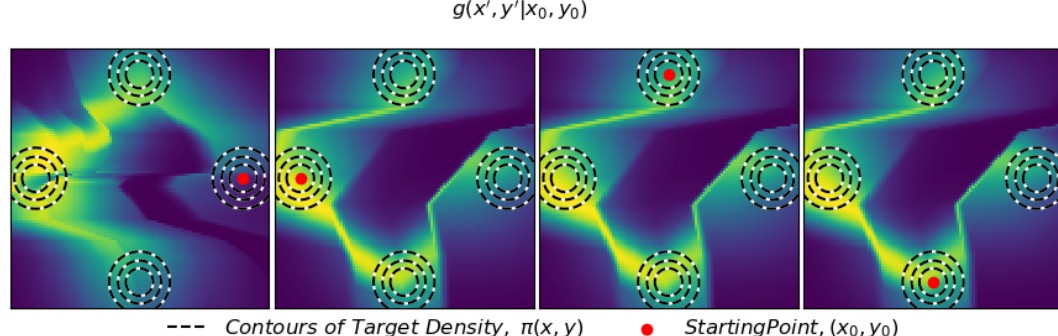

Figure 9: Comparison of multi-scheme mixture proposals (L=3) of Equation 4 obtained by optimizing GSM targeting an acceptance rate of 0.3. Each subplot illustrates the proposal distribution's density when sampling from a given starting position (in red).

$g(x', y'|x_0, y_0)$

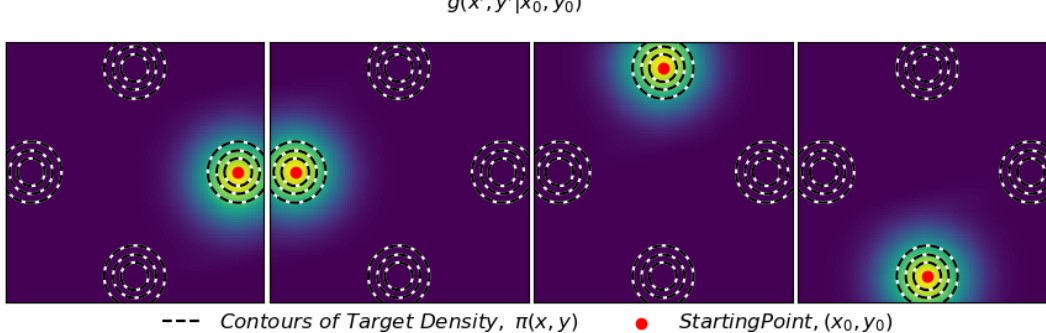

--- Contours of Target Density, $\pi(x, y)$     ● StartingPoint, $(x_0, y_0)$

Figure 10: Comparison of multi-scheme mixture proposals (L=3) of Equation 4 obtained by optimizing GSM targeting an acceptance rate of 0.6. Each subplot illustrates the proposal distribution's density when sampling from a given starting position (in red).

$g(x', y'|x_0, y_0)$

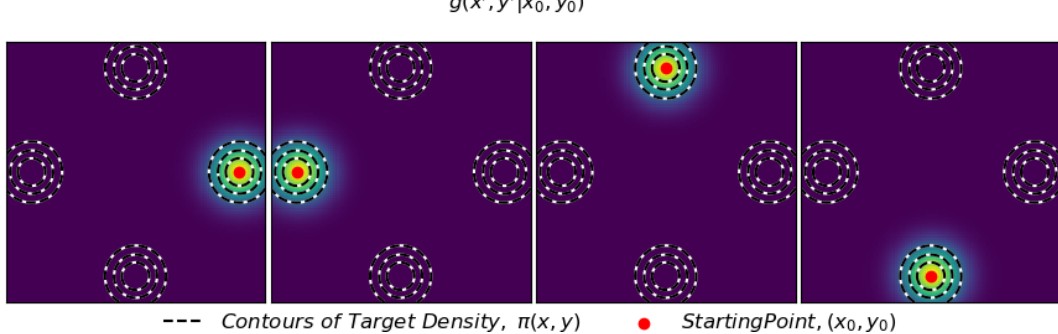

--- Contours of Target Density, $\pi(x, y)$     ● StartingPoint, $(x_0, y_0)$

Figure 11: Comparison of multi-scheme mixture proposals (L=3) of Equation 4 obtained by optimizing GSM targeting an acceptance rate of 0.9. Each subplot illustrates the proposal distribution's density when sampling from a given starting position (in red).

## H    FURTHER EXPLANATION OF MULTI-SCHEME OPTIMIZATION RESULTS

To provide an intuitive explanation of the results of Section 7.1, we will start by restating (in a more simplified form from the original in Section 3) the overall aim of MCMC proposal optimization, namely finding a solution to the optimization problem:

$$g_{opt} = \underset{g \in G}{\operatorname{argmin}} \, \mathcal{L}[g; \pi]$$

Where $G$ is the model class of our proposal architecture, $\pi$ is the target distribution, and $\mathcal{L}$ is the objective function chosen for the optimization. With the experimental setup of Section 7.1, we are confident that the results in Table 2 reflect proposals that are very close to the global optima of the compared objective functions. Figure 12 illustrates the placement of the global optima for MSJD within the model class of the multi-scheme proposal distribution of Equation 3 in a very abstract and simplified manner. For simplicity, we also plot hypothetical locations for RWM sampling and i.i.d. resampling from the target distribution, $\pi$.

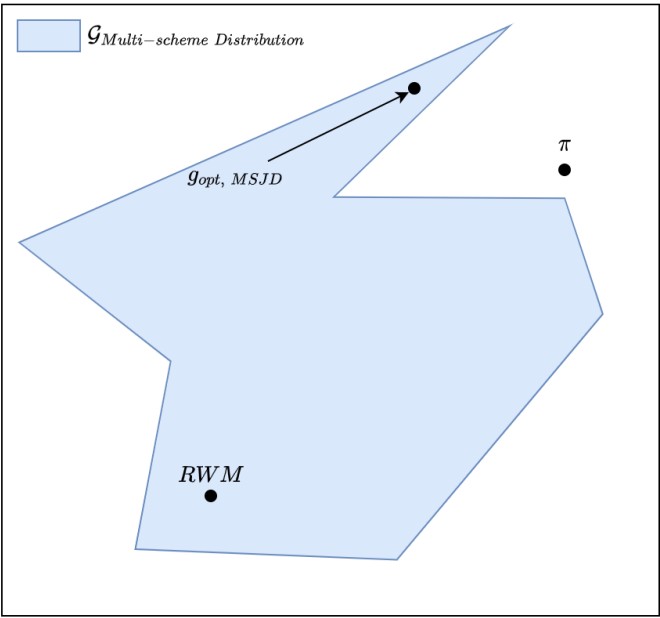

Figure 12: Location of the optimum of MSJD when optimizing the multi-scheme architecture of Equation 3.

Fundamentally, the highly expressive model class of the multi-scheme distribution offers many pathways to increase MSJD. The end result is, unfortunately, a pathologically non-ergodic Markov Chain. Even though MSJD is very well aligned with our notion of MCMC efficiency in model class neighborhoods very close to preconditioned RWM, MALA, and HMC, it is clear that MSJD and its derivatives are not robust measures of MCMC efficiency for proposal architectures with model classes like those of our example multi-scheme distribution.

When utilizing the GSM, the optimization procedure is complicated slightly by targeting the user-specified acceptance rate. Effectively, this restricts $G$ to those proposal distributions within the model class that attain the user specified acceptance rate. Figures 13 and 14 illustrate the placement of the global optima for the GSM when targeting acceptance rates of 30% and 60% using the multi-scheme

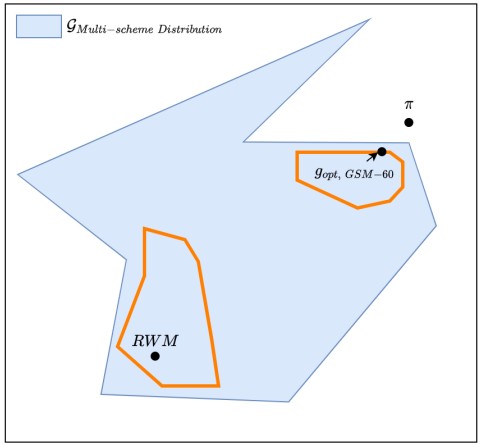 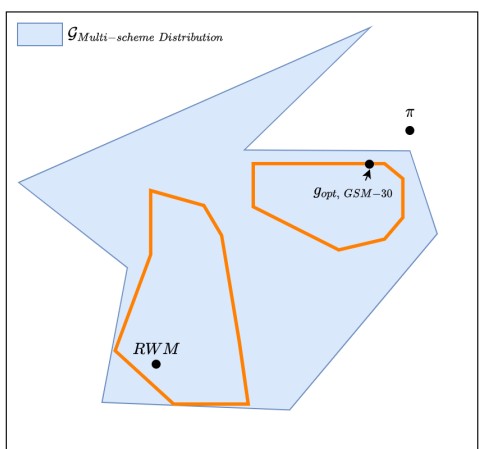

(a) Targeting acceptance rate of 60%.        (b) Targeting acceptance rate of 30%.

Figure 13: Locations of the optima of the GSM when optimizing the multis-scheme architecture of Equation 3 with 8 coupling layers targeting acceptance rates of (a) 60% and (b) 30%.Orange lines represent the restrictions of the model classes that attain the desired acceptance rates.

distribution of Equation 3 with 3 and 8 coupling layers. In these figures, the orange lines represent the restrictions of the model classes that attain the desired acceptance rates.

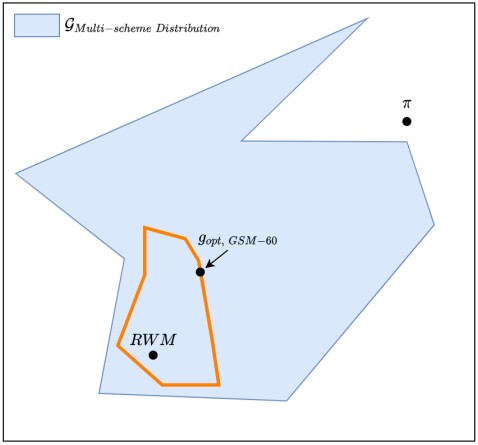 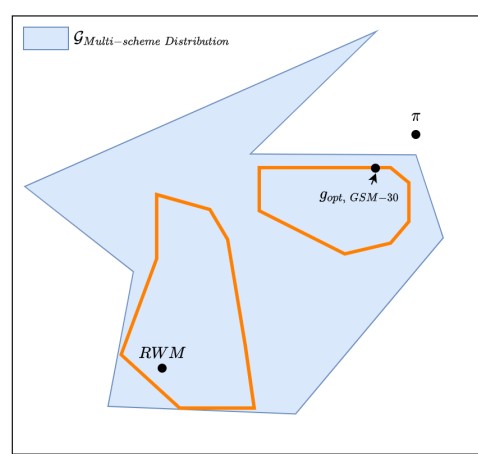

(a) Targeting acceptance rate of 60%.        (b) Targeting acceptance rate of 30%.

Figure 14: Locations of the optima of the GSM when optimizing the multi-scheme architecture of Equation 3 with 3 coupling layers targeting acceptance rates of (a) 60% and (b) 30%. Orange lines represent the restrictions of the model classes that attain the desired acceptance rates.

There are two model class regimes that are most relevant for the optimization of the multi-scheme architecture with the GSM in Section 7.1: a neighborhood of primarily inefficient, RWM-like proposals and a neighborhood of approximate i.i.d. resampling proposals. With 8 coupling layers, both

regimes are accessible while maintaining acceptance rates of 30% and 60%. In Figures Figures 13 (a) and (b), this is represented by the presence of two orange surfaces within $G_{Multi-scheme\ Distribution}$, one in the lower-left (representing the RWM-like regime) and one in the upper-right (representing the approximate resampling regime). When using 8 coupling layers, specifying target acceptance rates of either 30% or 60% will still include some portion of the approximate resampling regime of the multi-scheme architecture's model class within the restriction over which the GSM optimizes. In the end, with 8 coupling layers, GSM-30 and GSM-60 have global optima that lie within the approximate resampling regime of the multi-scheme architecture.

Moving on to the case with 3 coupling layers, the architecture's ability to approximate resampling is diminished, which greatly impacts the qualitative behaviour of the restrictions of the model class at different target acceptance rates. As depicted in Figure 14 (b), the case of optimizing a multi-scheme architecture with 3 coupling layers with the GSM targeting an acceptance rate of 30% is not significantly changed, as there is there is a portion of the approximate resampling regime that lies within the restriction of attaining an acceptance rate of 30% and the global optimum of GSM-30 is some point within this restriction of the approximate sampling regime. However, when using 3 coupling layers and targeting an acceptance rate of 60%, there is no member of the approximate sampling regime that falls into the restriction over which the GSM is optimized, leaving only inefficient RWM-like proposals from which to choose the optimum of GSM-60 in this case. Without significant prior knowledge regarding the interaction between the proposal architecture and the target distribution or a significant time investment testing many different acceptance rates, there is little that can be done to avoid the detrimental effect of choosing a highly sub-optimal target acceptance rate when optimizing model classes like the multi-scheme distribution using the GSM.

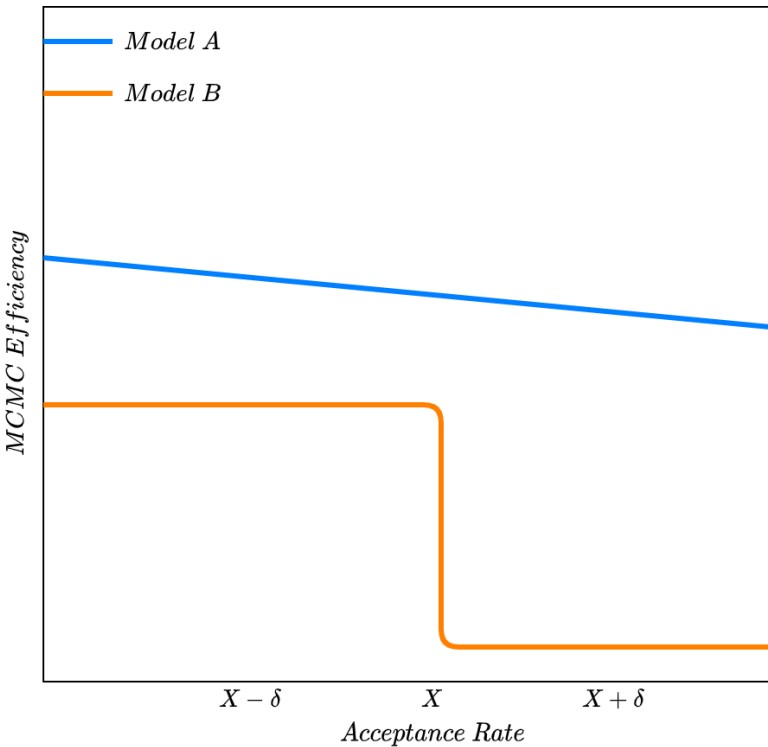

Figure 15: Comparison of maximal MCMC efficiency ($\mathcal{L}^*$) attained given a target acceptance rate in the neighborhood of accepance rate $X$. Model A exhibits continuous behaviour, as we might expect from restricted architectures like preconditioned RWM. Model B exhibits discontinuous behaviour, as exhibited by the multi-scheme architecture with 3 coupling layers around an acceptance rate of 55-56%.

The "drop-out" of the approximate resampling regime when using 3 coupling layers with the multi-scheme distribution and targeting an acceptance rate of 60% is reflective of a discontinuity of MCMC

efficiency with respect to acceptance rate that should be expected of expressive neural MCMC architectures like the multi-scheme proposal distribution. With very restricted architectures, like preconditioned RWM or MALA, our notion of maximal MCMC efficiency attained at a given acceptance rate remains continuous with respect to the acceptance rate. In the nomenclature of Section 4, $\mathcal{L}^*$ is very tame when optimized under the restriction of attaining a fixed acceptance rate with the model classes of preconditioned RWM and MALA. So when we find RWM optimized to an acceptance rate of 20% or MALA optimized to an acceptance rate of 60%, we are reasonably confident that the proposals are close to optimality and are not drastically inefficient. But, when using a proposal architecture like the multi-scheme distribution, $\mathcal{L}^*$ can behave very sharply with respect to the target acceptance rate.

This is illustrated in Figure 15, where the maximal MCMC efficiency ($\mathcal{L}^*$) at a target acceptance rate is plotted against the target acceptance rate for two proposal architectures (A and B). In the neighborhood of acceptance rate $X$, Model A (corresponding to the multi-scheme architecture with 8 coupling layers above) behaves much as we would expect given our experiences with RWM and MALA, where minor errors in fixing a target acceptance rate have minor impacts on the end efficiency. For Model B (corresponding to the multi-scheme architecture with 3 coupling layers above), the interaction of the proposal architecture and the target distribution results in a discontinuity in optimized MCMC efficiency at a target acceptance rate $X$ (in our case, corresponding to around 55-56%). Here, very minor differences in the target acceptance rate can result in an orders of magnitude change in MCMC efficiency for Model B. Models A and B in this example do not need to differ by their proposal architecture, but may instead differ by their target distribution, which makes it difficult to translate the knowledge gained from successful optimization in one task to the next task with the same proposal architecture.

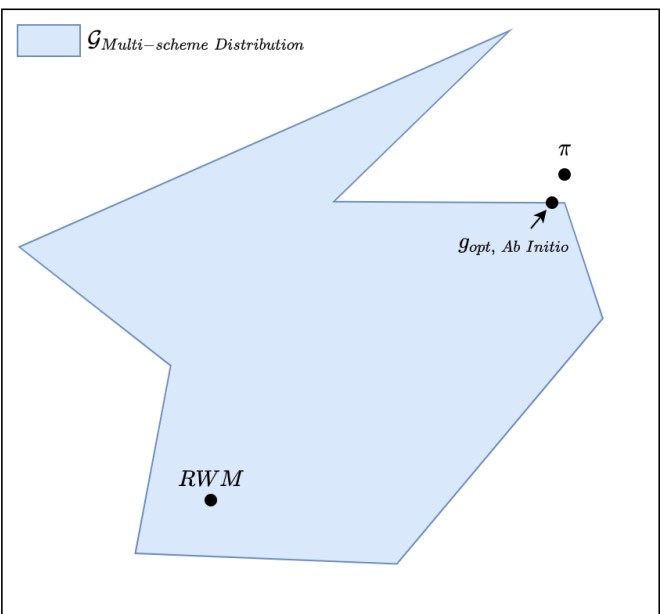

Figure 16: Location of the optimum of an Ab Initio objective when optimizing the multi-scheme architecture of Equation 3.

As robust approximations of our notion of MCMC efficiency throughout the expansive model classes of architectures like the multi-scheme distribution, Ab Initio objective functions allow us, in principle, to confidently approach MCMC proposal optimization with highly expressive neural proposals as

we would any other traditional deep learning task. Figure 16 illustrates the placement of the global optima for an Ab Initio objective function within the model class of the multi-scheme proposal distribution of Equation 3. In contrast to MSJD based objectives, the functional constraints of Section 4 and the fitting and verification procedures of Section 6 work to ensure that Ab Initio objective functions remain robust and align with our notion of MCMC efficiency throughout the model classes of very expressive architectures. In contrast to the GSM, Ab Initio objective functions seek to directly approximate our notion of MCMC efficiency without relying on prior knowledge of optimal acceptance rates, which are very difficult to determine for arbitrary neural proposal architectures and target distributions. The overall result is that Ab Initio objective functions offer global optima for highly expressive neural proposal architectures that remain well aligned with our notion of MCMC efficiency and are compatible with a traditional deep learning approach to optimizing neural architectures.

## I   POSITION-DEPENDENT MIXTURE PROPOSAL OPTIMIZATION

To examine general objective function behavior without additional complications arising from particular proposal architectures, we first consider the optimization of proposals consisting of a mixture of gaussians (with position independent means and covariances) with position dependent component weights specified by a neural network. This proposal distribution may be specified as follows:

$$\begin{aligned} \vec{\mathbf{x}}'|\vec{\mathbf{x}}, (S = i) &\sim \mathcal{N}\{\mu_{i,\theta}; \Sigma_{i,\theta}\} \\ P(S = i|\vec{\mathbf{x}}) &= f_\theta(\vec{\mathbf{x}}) \end{aligned} \tag{4}$$

The target distribution is an equal mixture of 4 standard gaussians positioned in a cross formation with a maximal distance of 8 between component centers. The proposal distribution consists of 4 gaussian components. The position-dependent weights, $f_\theta(\vec{\mathbf{x}})$, are specified by a standard ReLU network (input dimension 2, hidden dimension 16, 4 hidden layers, output dimension 4) followed by a component-wise softmax layer. Proposals are initialized near i.i.d. sampling of the target. For optimization, 40000 gradient steps are taken, with each training batch taken from a single starting point sampled i.i.d. from the target distribution from which 50 independent proposals from $g_\theta(x'|x)$ are generated to empirically estimate expected loss on the basis of 50 total samples. The gradient update steps take the form of Algorithm 1 with $M = 1$ and $N = 50$. Optimization is performed using the Adam optimizer (Kingma and Ba, 2015) with a stepsize of 3e-4. For optimizing the GSM (Titsias and Dellaportas, 2019), $\beta$ is initially set to $0.5$ and $\rho_\beta$ is set to $0.002$. After optimization, acceptance rate, MSJD, and ESS are estimated for each optimized model based on 5 independent Markov Chains of 1000 proposals with starting points independently sampled from the target distribution. To gather statistics regarding the mean and standard error of reported variables, each optimization is replicated a total 5 times. Figures 17, 18, 19, 20, 21, and 22 show density plots of the position-dependent weighting functions obtained following optimization using the Ab Initio objective function of Equation (2), MSJD (Pasarica and Gelman, 2010), L2HMC's objective (Levy et al., 2018) and the GSM (Titsias and Dellaportas, 2019) with varying target acceptance rates.

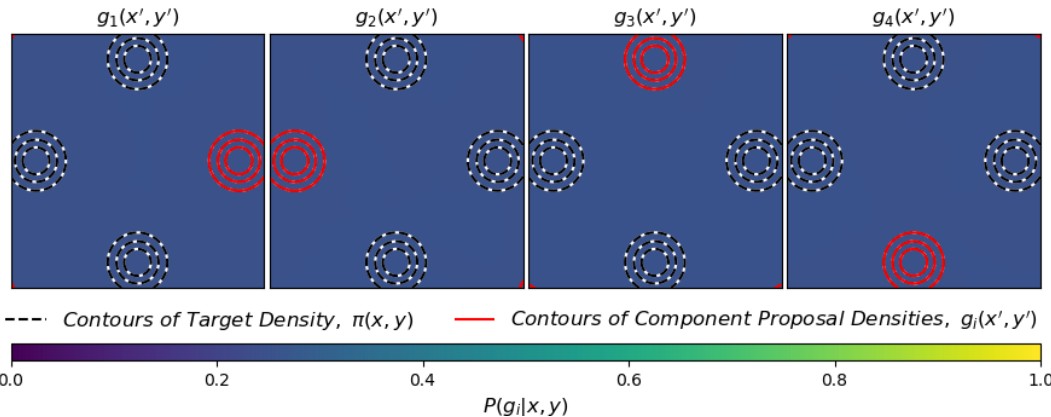

Figure 17: Comparison of position-dependent mixture proposals of Equation 3 obtained by optimizing Equation (2). Each subplot illustrates the contours of a proposal component (in red) alongside a density plot of the probability of sampling from that component based on starting position.

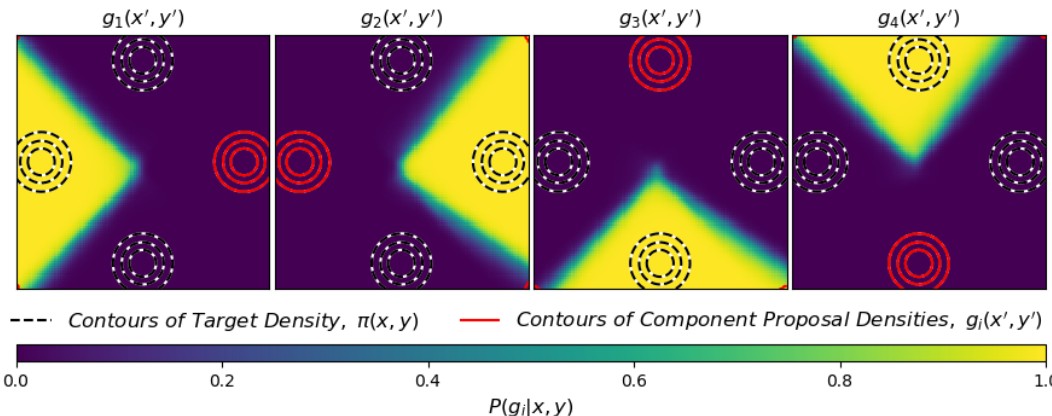

Figure 18: Comparison of position-dependent mixture proposals of Equation 3 obtained by optimizing MSJD. Each subplot illustrates the contours of a proposal component (in red) alongside a density plot of the probability of sampling from that component based on starting position.

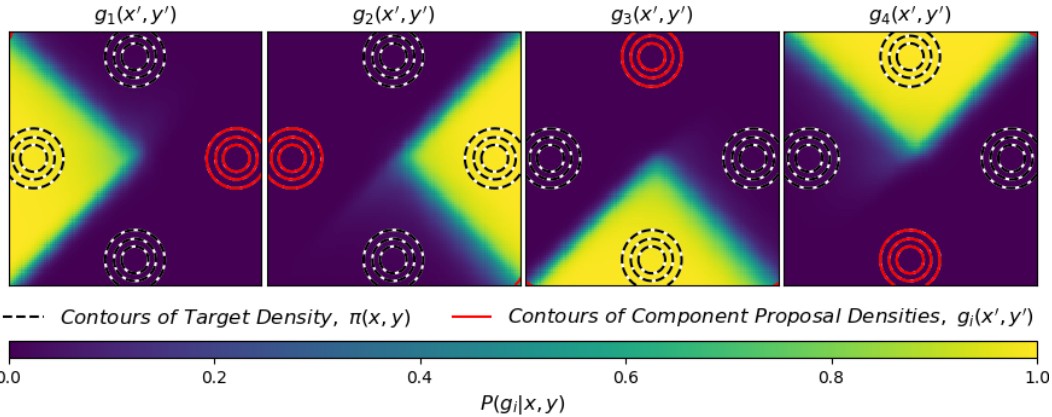

Figure 19: Comparison of position-dependent mixture proposals of Equation 3 obtained by optimizing L2HMC's objective. Each subplot illustrates the contours of a proposal component (in red) alongside a density plot of the probability of sampling from that component based on starting position.

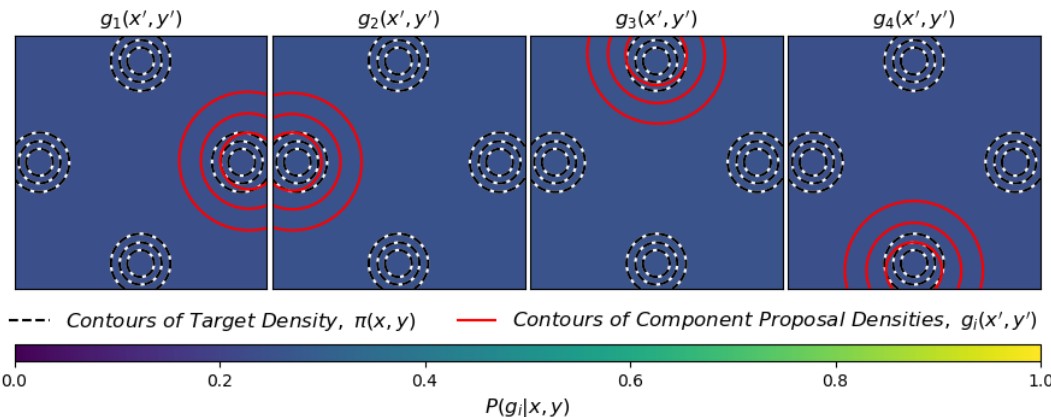

Figure 20: Comparison of position-dependent mixture proposals of Equation 3 obtained by optimizing GSM targeting an acceptance rate of 0.3. Each subplot illustrates the contours of a proposal component (in red) alongside a density plot of the probability of sampling from that component based on starting position.

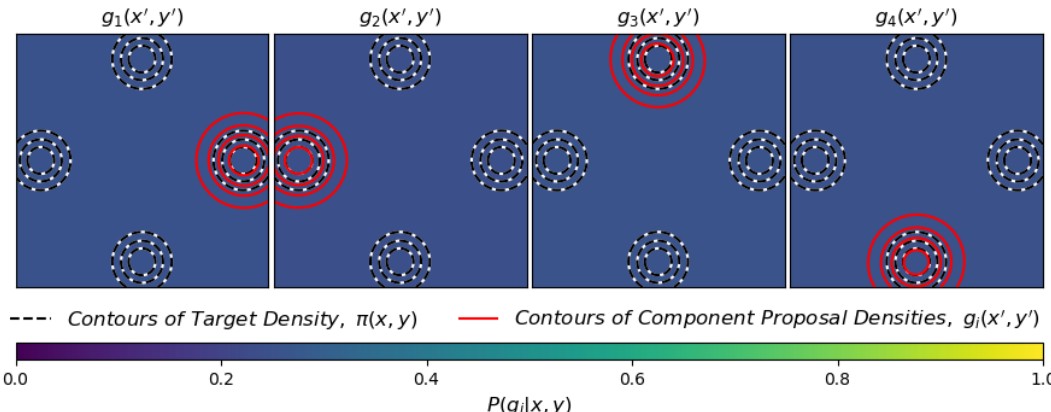

Figure 21: Comparison of position-dependent mixture proposals of Equation 3 obtained by optimizing GSM targeting an acceptance rate of 0.6. Each subplot illustrates the contours of a proposal component (in red) alongside a density plot of the probability of sampling from that component based on starting position.

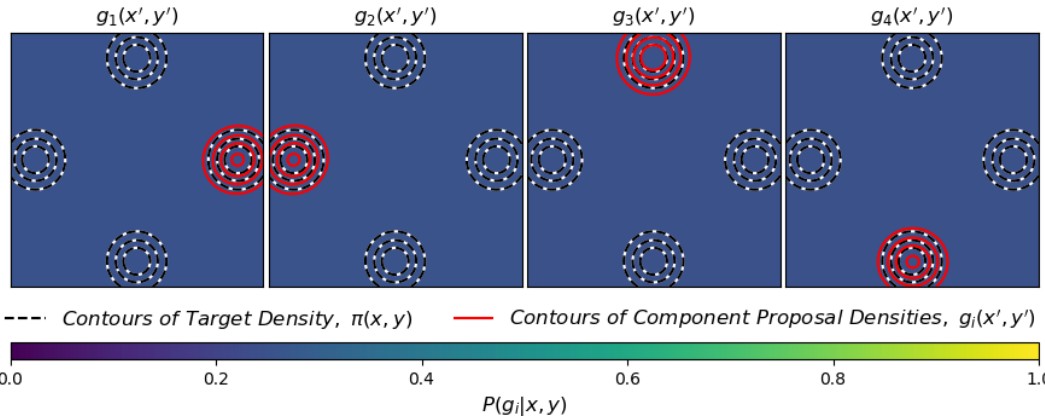

Figure 22: Comparison of position-dependent mixture proposals of Equation 3 obtained by optimizing GSM targeting an acceptance rate of 0.9. Each subplot illustrates the contours of a proposal component (in red) alongside a density plot of the probability of sampling from that component based on starting position.

The results of these optimizations are provided in Table 12.

Table 12: Comparison of sampling performance obtained optimizing the position-dependent mixture proposals of Equation (4) using various objective functions. Value means are reported to at most the first significant digit of standard error (reported in parentheses).

| | Acc. Rate | MSJD | ESS per Proposal | | Grad. Per Sec. |
| --- | --- | --- | --- | --- | --- |
| | | | $x_i$ | $x_i^2$ | |
| Ab Initio, Eq. (2) | 0.998(1) | 34.0(3) | 1.00(2) | 1.05(3) | 17.3(3) |
| | | | | | |
| MSJD Opt. | 0.998(1) | 65.9(4) | 1.00(1) | 6(1)e-4 | 24(2) |
| L2HMC Obj. | 0.998(1) | 66.0(2) | 1.01(1) | 7(1)e-4 | 23(1) |
| | | | | | |
| GSM-90 | 0.900(2) | 30.8(3) | 0.81(2) | 0.81(6) | |
| GSM-60 | 0.601(5) | 20.9(2) | 0.41(2) | 0.40(4) | 20(1) |
| GSM-30 | 0.297(7) | 10.3(4) | 0.15(1) | 0.16(1) | |
| | | | | | |
| I.I.D. Resample | 1.000 | 34.2(2) | 0.96(5) | 1.01(3) | |

We find that i.i.d. resampling remains a stable global minimum of our Ab Initio objective function. The GSM objective also optimizes towards a constant and equal weighting of the proposals components, but increases the variance of each component to match its targeted acceptance rate. Both MSJD and L2HMC's modification find global minima at arbitrarily non-ergodic proposals. Both of these MSJD based objectives result in Markov Chains that only effectively explore half of the target's probability mass (either the horizontal or vertical components), which is numerically reflected by the poor ESS performance for estimating second moments.

## J  FURTHER DETAILS FOR LOGISTIC REGRESSION POSTERIOR SAMPLING

For this experiment, we augment the target distribution with a number of auxiliary variables. Similar to the momenta in HMC, these augmenting variables are independent of the original data variables and are distributed following a standard multivariate gaussian. Proposal distributions are optimized to target the joint distribution of original and augmenting variables. For sampling, the augmenting variables are resampled before making each proposal, similar to the procedure employed in HMC. With one exception (the Ripley dataset) we found best results by using a number of augmenting variables, $a$, equal to twice the dimensionality of the original data, $d$.

This experiment again uses multi-scheme proposals, specified as follows:

$$\vec{\mathbf{n}} \sim \mathcal{N}\{\vec{\mathbf{0}} \, ; \, I\}$$
$$\vec{\mathbf{z}}'|\vec{\mathbf{x}}, \vec{\mathbf{n}} = \mu_{L,\theta}(T_\theta^{-1}(\vec{\mathbf{x}})) + \Sigma_{L,\theta}(T_\theta^{-1}(\vec{\mathbf{x}})) \odot \vec{\mathbf{n}}$$
$$\vec{\mathbf{x}}' = \mu_{D,\theta}(\vec{\mathbf{x}}) + \Sigma_{D,\theta}(\vec{\mathbf{x}}) \odot T_\theta(\vec{\mathbf{z}}')$$

The target distribution is the posterior distribution of regression weights for logistic regression of various datasets from the UCI repository (Bache and Lichman, 2013) and from Ripley (2007). The prior distributions used for regression weights are independent gaussians with a mean of 0 and a standard deviation of 10. The additive functions, $\mu_{L,\theta}$ and $\mu_{D,\theta}$, are specified by standard ReLU networks (input dimension $a + d$, hidden dimension (W) $3(a+d)$, 4 hidden layers (D), output dimension $a + d$). The multiplicative functions, $\Sigma_{L,\theta}$ and $\Sigma_{D,\theta}$, are specified by standard ReLU networks (input dimension $a + d$, hidden dimension (W) $3(a+d)$, 4 hidden layers (D), output dimension $a + d$) followed by component-wise exponentiation. The normalizing flow, $T_\theta$, follows the NICE architecture (Dinh et al., 2014), with additive coupling layers specified by standard ReLU networks (input dimension $\frac{a+d}{2}$, hidden dimension (W) $3(a+d)$, 4 hidden layers (D), output dimension $\frac{a+d}{2}$). For this experiment, the same number of coupling layers is used ($L = 5$) for all datasets. Table 13 summarizes the network parameters used in this experiment.

Before each optimization, a long equilibrated Markov Chain is obtained following 500,000 proposals of HMC tuned to attain an acceptance rate of 0.65 (using hamiltorch's (Cobb et al., 2019) implementation of HMC). A burn-in period of 1000 proposals is utilized for this HMC tuning and is discarded.

For optimization, 40000 gradient steps are taken, with each training batch taken from 8 starting points sampled randomly from HMC Markov Chain and from each starting point 100 independent proposals from $g_\theta(x'|x)$ are generated to empirically estimate expected loss on the basis of 800 total samples. The gradient update steps take the form of Algorithm 1 with $M = 8$ and $N = 100$. Uniform sampling from this Markov Chain approximates i.i.d sampling from the target. Optimization is performed using the Adam optimizer (Kingma and Ba, 2015) with a stepsize of 3e-4. For optimizing GSM (Titsias and Dellaportas, 2019), $\beta$ is initially set to $d^{-1}$ and $\rho_\beta$ is set to $0.02$ and initialization is varied to ensure the target acceptance rate is attained. After optimization, acceptance rate, MSJD, and ESS are calculated solely based on original data dimensions and are estimated for each optimized model based on 5 independent Markov Chains of 20000 proposals with starting points independently sampled from the target distribution. To gather statistics regarding the mean and standard error of reported variables, each optimization is replicated a total 5 times. Tables 14, 15, 16, 17, and 18 summarize the sampling performance measures obtained following optimization, including minimum, median, and maximum one-dimensional ESS obtained across all dimensions of the original data variables. Minimum one-dimensional ESS is most indicative of sampling efficiency across the entire state space of the distribution and is therefore the focus of our comparisons.

Table 13: Specification of network parameters used in multi-scheme proposals for logistic regression sampling tasks.

| | a | d | $\mu_{L,\theta}$ and $\mu_{D,\theta}$ | | $\Sigma_{L,\theta}$ and $\Sigma_{D,\theta}$ | | $T_\theta$ | | |
|---|---|---|---|---|---|---|---|---|---|
| | | | W | D | W | D | W | D | L |
| German Credit | 21 | 21 | 126 | 4 | 126 | 4 | 126 | 4 | 5 |
| Australian Credit | 15 | 15 | 90 | 4 | 90 | 4 | 90 | 4 | 5 |
| Heart | 14 | 14 | 84 | 4 | 84 | 4 | 84 | 4 | 5 |
| Pima | 9 | 9 | 54 | 4 | 54 | 4 | 54 | 4 | 5 |
| Ripley | 6 | 3 | 27 | 4 | 27 | 4 | 27 | 4 | 5 |

Table 14: Results from optimizing multi-scheme proposal distributions targeting posterior distribution of logistic regression weights for German Credit dataset. Value means are reported to at most the first significant digit of standard error (reported in parentheses).

| | Acc. Rate | MSJD | ESS per Proposal $x_i$ | $x_i^2$ | Grad. Per Sec. |
|---|---|---|---|---|---|
| Ab Initio | 0.81(1) | 0.30(1) | 0.56(3) | 0.49(7) | 11.5(6) |
| MSJD Opt. | 0.77(2) | 0.32(1) | 0.52(7) | 0.4(1) | 15(1) |
| L2HMC Obj. | 0.76(2) | 0.32(1) | 0.54(4) | 0.48(5) | 14.8(9) |
| GSM-90 | 0.90(1) | 0.012(1) | 0.005(1) | 0.005(1) | |
| GSM-60 | 0.61(3) | 0.24(1) | 0.40(3) | 0.40(3) | 10.1(3) |
| GSM-30 | 0.29(5) | 0.13(2) | 0.13(4) | 0.13(4) | |
| HMC | 0.65 | 0.201(3) | 0.006(3) | 0.005(3) | |
| I.I.D. Resample | 1.00 | 0.3700(3) | 0.98(1) | 0.983(3) | |

Table 15: Results from optimizing multi-scheme proposal distributions targeting posterior distribution of logistic regression weights for Australian Credit dataset. Value means are reported to at most the first significant digit of standard error (reported in parentheses).

| | Acc. Rate | MSJD | ESS per Proposal | | Grad. Per Sec. |
| --- | --- | --- | --- | --- | --- |
| | | | $x_i$ | $x_i^2$ | |
| Ab Initio | 0.85(1) | 1.58(4) | 0.63(4) | 0.57(8) | 12.2(2) |
| MSJD Opt. | 0.77(2) | 2.39(3) | 0.55(2) | 0.53(3) | 14(1) |
| L2HMC Obj. | 0.76(1) | 2.39(3) | 0.47(4) | 0.41(7) | 14.5(7) |
| GSM-90 | 0.901(4) | 0.088(3) | 0.0105(2) | 0.0105(2) | |
| GSM-60 | 0.61(1) | 1.23(2) | 0.41(1) | 0.42(1) | 10.6(2) |
| GSM-30 | 0.32(1) | 0.71(3) | 0.16(1) | 0.16(1) | |
| HMC | 0.65 | 1.43(2) | 0.02(1) | 0.02(1) | |
| I.I.D. Resample | 1.00 | 1.901(4) | 0.97(1) | 0.97(1) | |

Table 16: Results from optimizing multi-scheme proposal distributions targeting posterior distribution of logistic regression weights for Heart dataset. Value means are reported to at most the first significant digit of standard error (reported in parentheses).

| | Acc. Rate | MSJD | ESS per Proposal | | Grad. Per Sec. |
| --- | --- | --- | --- | --- | --- |
| | | | $x_i$ | $x_i^2$ | |
| Ab Initio | 0.86(2) | 1.03(4) | 0.63(6) | 0.58(7) | 11.7(4) |
| MSJD Opt. | 0.81(1) | 1.40(3) | 0.69(3) | 0.32(6) | 15.0(3) |
| L2HMC Obj. | 0.82(1) | 1.43(4) | 0.73(4) | 0.28(4) | 14.7(7) |
| GSM-90 | 0.900(3) | 0.138(3) | 0.034(2) | 0.035(1) | |
| GSM-60 | 0.59(2) | 0.72(2) | 0.34(3) | 0.35(3) | 10.8(3) |
| GSM-30 | 0.29(3) | 0.42(4) | 0.13(2) | 0.14(2) | |
| HMC | 0.65 | 0.70(1) | 0.02(1) | 0.02(1) | |
| I.I.D. Resample | 1.00 | 1.222(1) | 0.97(1) | 0.97(2) | |

Table 17: Results from optimizing multi-scheme proposal distributions targeting posterior distribution of logistic regression weights for Pima dataset. Value means are reported to at most the first significant digit of standard error (reported in parentheses).

| | Acc. Rate | MSJD | ESS per Proposal | | Grad. Per Sec. |
| --- | --- | --- | --- | --- | --- |
| | | | $x_i$ | $x_i^2$ | |
| Ab Initio | 0.88(3) | 0.18(1) | 0.7(1) | 0.7(1) | 11.2(8) |
| MSJD Opt. | 0.84(2) | 0.28(1) | 0.79(6) | 0.13(6) | 15.0(9) |
| L2HMC Obj. | 0.84(4) | 0.27(1) | 0.78(5) | 0.14(8) | 15.2(7) |
| GSM-90 | 0.91(1) | 0.186(3) | 0.69(6) | 0.68(6) | |
| GSM-60 | 0.60(3) | 0.14(1) | 0.43(3) | 0.44(3) | 11.1(5) |
| GSM-30 | 0.30(3) | 0.08(1) | 0.15(2) | 0.15(2) | |
| HMC | 0.65 | 0.139(3) | 0.007(2) | 0.007(2) | |
| I.I.D. Resample | 1.00 | 0.2118(2) | 0.98(1) | 0.98(1) | |

Table 18: Results from optimizing multi-scheme proposal distributions targeting posterior distribution of logistic regression weights for Ripley dataset. Value means are reported to at most the first significant digit of standard error (reported in parentheses).

| | Acc. Rate | MSJD | ESS per Proposal | | Grad. Per Sec. |
| --- | --- | --- | --- | --- | --- |
| | | | $x_i$ | $x_i^2$ | |
| Ab Initio | 0.97(1) | 0.48(1) | 0.90(4) | 0.88(4) | 11.6(7) |
| MSJD Opt. | 0.92(1) | 0.87(2) | 1.35(6) | 1.2(1) | 14.3(8) |
| L2HMC Obj. | 0.91(2) | 0.87(3) | 1.4(1) | 1.2(1) | 15.1(9) |
| GSM-90 | 0.90(1) | 0.46(1) | 0.82(3) | 0.81(3) | |
| GSM-60 | 0.61(2) | 0.35(1) | 0.50(1) | 0.51(2) | 11.3(6) |
| GSM-30 | 0.30(1) | 0.20(1) | 0.20(1) | 0.20(1) | |
| HMC | 0.65 | 0.202(5) | 0.15(3) | 0.18(3) | |
| I.I.D. Resample | 1.00 | 0.498(2) | 1.00(1) | 0.99(2) | |

The results for the Heart, Pima, and Ripley datasets exhibit the same behavior observed within the experiments involving the gaussian mixture target in Section 7.1. Optimizing MSJD or L2HMC's objective yields proposals whose behavior significantly deviates from i.i.d. resampling from the target distribution and result in a proposal MSJD much larger than what would be obtained following i.d.d. resampling from the target. As seen in Tables 16 and 17, the target distributions with the Heart and Pima datasets are similar to those with the gaussian mixture target in that this behavior is numerically evidenced by a severe decrease in ESS relating to estimation of the distribution's second moments. From Table 18, we see that the target distribution with the Ripley dataset is such that this behavior is numerically evidenced by ESS calculations significantly greater than 1, which is indicative of the proposal distribution yielding greatly negative auto-correlations. These results demonstrate that this behavior is not the result of specific choices in proposal architecture or target distribution. We conclude that this sub-optimal behavior when maximizing MSJD or optimizing L2HMC's objective occur when the proposal distribution allows sufficiently complex (relative to the target distribution) position dependence.

## K   FURTHER DETAILS FOR SCHEME SELECTION EXPERIMENT

For this experiment, the target distribution is the posterior distribution of regression weights for logistic regression of 5's and 6's from the training set of MNIST (LeCun et al., 1998) digits (d=785). The prior distributions used for regression weights are independent gaussians with a mean of 0 and a standard deviation of 10. Our proposal distributions are diagonally preconditioned MALA, diagonally preconditioned RWM, the multi-scheme proposal distribution of Equation (3), and a normalizing flow intended to solely approximate i.i.d. resampling from the target.

Before each optimization, a long equilibrated Markov Chain is obtained following 500,000 proposals of HMC tuned to attain an acceptance rate of 0.65 (using hamiltorch's (Cobb et al., 2019) implementation of HMC). A burn-in period of 1000 proposals is utilized for this HMC tuning and is discarded. For optimization, 40000 gradient steps are taken, with each training batch taken from 1 starting point sampled randomly from HMC Markov Chain and from each starting point 1 proposal $g_\theta(x'|x)$ is generated to empirically estimate expected loss. The gradient update steps take the form of Algorithm 1 with $M = 1$ and $N = 1$. Uniform sampling from this Markov Chain approximates i.i.d sampling from the target. Optimization is performed using the Adam optimizer (Kingma and Ba, 2015). After the initial 40000 steps, the Ab Initio objective function of Equation (2) is estimated following 10000 proposals from starting points independently sampled from the long equlibrated HMC chain. Our rough, upper-bound estimates for the MSJD and acceptance rates of the resampling normalizing were estimated using the same procedure as with the Ab Initio objective function.

For optimizing both the preconditioned MALA and RWM proposal distributions, a stepsize of 3e-4 is used. One of the preconditioned MALA distributions and the preconditioned RWM distribution are

optimized using the Ab Initio objective function of Equation (2). One of the preconditioned MALA distributions and is optimized to maximize MSJD.

For the multi-scheme proposal distribution, the additive functions, $\mu_{L,\theta}$ and $\mu_{D,\theta}$, are specified by standard ReLU networks (input dimension 785, hidden dimension (W) 2355, 4 hidden layers (D), output dimension 785). The multiplicative functions, $\Sigma_{L,\theta}$ and $\Sigma_{D,\theta}$, are specified by standard ReLU networks (input dimension 785, hidden dimension (W) 2355, 4 hidden layers (D), output dimension 785) followed by component-wise exponentiation. The normalizing flow, $T_\theta$, follows the NICE architecture (Dinh et al., 2014), with additive coupling layers specified by standard ReLU networks (input dimension 392, hidden dimension (W) 2355, 4 hidden layers (D), output dimension 392). The flow uses 5 coupling layers A stepsize of 3e-5 is used. The multi-scheme proposal distribution is optimized using the Ab Initio objective function of Equation (2).

The resampling normalizing flow follows the NICE architecture (Dinh et al., 2014), with additive coupling layers specified by standard ReLU networks (input dimension 392, hidden dimension (W) 3140, 5 hidden layers (D), output dimension 392). The flow uses 6 coupling layers A stepsize of 3e-6 is used. The resampling normalizing flow is optimized to optimize negative log-likelihood.

After optimization, acceptance rate, MSJD, and ESS are calculated solely based on original data dimensions and are estimated for each optimized model based on 100 independent Markov Chains of 1000 proposals with starting points independently sampled from the target distribution. To gather statistics regarding the mean and standard error of reported variables, each optimization is replicated a total 5 times.

## L    CALCULATING AND EVALUATING EFFECTIVE SAMPLE SIZE

Throughout our experiments, we use ESS as a means of verifying the sampling performance of optimized proposal distributions. From a series of samples drawn sequentially from a Markov Chain, $(X_1, X_2, ..., X_N)$, the simplest estimation of one-dimensional ESS can be obtained by the single chain calculation:

$$ESS = \frac{N}{1 + 2 \sum_{t=1}^{\infty} \rho_t}$$

Where $\rho_t$ is the autocorrelation at lag $t$ within the series $(X_1, X_2, ..., X_N)$. For slowly mixing Markov Chains, more accurate estimations of ESS can be obtained by using multi-chain formulations that utilize multiple series of samples and considers both within-chain and between-chain variance (Vehtari et al. (2019) provide a summary of these multi-chain methods). In our experiments, we utilize PyMC's (Patil et al., 2010) calculation of ESS, which considers both within-chain and between-chain variance.

Although ESS is often viewed as providing a measure of how many effectively independent samples are provided within the series $(X_1, X_2, ..., X_N)$, it is important to keep in mind that this interpretation applies only for the purpose of estimating the particular expectation $\mathbb{E}[X]$. In general, calculating ESS from the series $(f(X_1), f(X_2), ..., f(X_N))$ relates to the uncertainty of estimating $\mathbb{E}[f(X)]$ based on the series $(X_1, X_2, ..., X_N)$. As these ESS results can vary dramatically depending on the expectation of interest, $\mathbb{E}[f(X)]$, a single ESS estimate does not completely summarize the ergodicity of an MCMC sampler for use in general calculations. For our evaluations, we consider ESS relating to estimating first and second moments of the target distribution ($\mathbb{E}[X]$ and $\mathbb{E}[X^2]$).

When evaluating the performance of restricted model class MCMC propsoals like RWM or MALA, ESS relating to estimating first moments ($\mathbb{E}[X]$) is often a reliable performance comparison and ESS per proposal results are typically found to be less than 1. Our results in Section 7.1 show that these expectations will not generally hold true when evaluating the performance of very expressive proposal distributions. As seen with proposals optimized with MSJD and L2HMC's objective, very expressive neural proposals can exhibit significant negative autocorrelations that yield ESS per proposals greater than 1 and can have first moment ESS results that mask significant non-ergodicities of the evaluated proposal distribution. When evaluating the performance of highly expressive neural MCMC proposals, additional consideration is required regarding what ESS truly measures and we cannot solely rely on the intuitions built from the applications of ESS to MCMC proposal distributions with very restricted model classes.

## M   OPTIMIZING MCMC OBJECTIVE FUNCTIONS AND ONLINE ADAPTATION

Throughout our experiments, we utilize either exact or approximate samples from $\pi(\vec{\mathbf{x}})$ to approximate the outer expectations ($\mathbb{E}_{\vec{\mathbf{x}} \sim \pi(\vec{\mathbf{x}})}[\ldots]$) found within all compared objective functions during optimization. As a result, our gradient update steps take the form illustrated in Algorithm 1 (note, when direct sampling from $\pi(\vec{\mathbf{x}})$ is not available, we approximate this sampling by uniformly sampling from a long equilibrated HMC chain). Algorithm 1 requires samples from $\pi(\vec{\mathbf{x}})$ to be available during optimization, which is typically not the case during practical applications of MCMC proposal optimization. For practical applications where samples from $\pi(\vec{\mathbf{x}})$ are not available, online adaptation procedures are used to approximate the outer expectations ($\mathbb{E}_{\vec{\mathbf{x}} \sim \pi(\vec{\mathbf{x}})}[\ldots]$), with gradient update steps taking a form like that illustrated in Algorithm 2. We must emphasize that we are not advocating for the use of update steps like Algorithm 1 in practice. We are instead using Algorithm 1 specifically because it allows us to most directly answer the core question of this work: determining whether the compared objective functions have optima that are aligned with our notion of MCMC efficiency when optimizing proposal distributions with highly expressive model classes like that of Equation (3).

---

**Algorithm 1:** Gradient Update Step with Sampling from $\pi(\vec{\mathbf{x}})$.

---

**Input:** Target distribution $\pi(\vec{\mathbf{x}})$, proposal distribution $g_\theta(\vec{\mathbf{x}}'|\vec{\mathbf{x}})$, objective function
$\qquad \mathcal{L}[\pi; g_\theta](\vec{\mathbf{x}}, \vec{\mathbf{x}}')$, integers $M, N \geq 1$, optimizer **ParamOpt**$(\hat{\mathcal{L}}, \theta)$.

**for** $i$ *in* **range**$(0, M)$ **do**
$\quad$ Sample $\vec{\mathbf{x}}_i \sim \pi(\vec{\mathbf{x}})$;
$\quad$ **for** $j$ *in* **range**$(0, N)$ **do**
$\quad\quad$ Sample $\vec{\mathbf{x}}'_j \sim g_\theta(\vec{\mathbf{x}}'|\vec{\mathbf{x}}_i)$;
$\quad$ **end**
**end**
$\hat{\mathcal{L}} \leftarrow \frac{1}{MN} \sum_{i,j} \mathcal{L}[\pi; g_\theta](\vec{\mathbf{x}}_i, \vec{\mathbf{x}}'_j)$;
$\theta \leftarrow$ **ParamOpt**$(\hat{\mathcal{L}}, \theta)$;

---

For the purposes of our experiments, we do not employ online adaptation because the adaptation procedures introduce an additional confounding factor to the analysis of performance differences between objective functions. By using Algorithm 1, our experiments demonstrate performance differences between the compared objective functions that can be directly attributed to the objective functions themselves and their interactions with the model classes of highly expressive proposal

---

**Algorithm 2:** Gradient Update Step with Online Adaptation.

---

**Input:** Target distribution $\pi(\vec{\mathbf{x}})$, proposal distribution $g_\theta(\vec{\mathbf{x}}'|\vec{\mathbf{x}})$, objective function
$\qquad \mathcal{L}[\pi; g_\theta](\vec{\mathbf{x}}, \vec{\mathbf{x}}')$, integers $M, N \geq 1$, $M$ initial Markov Chain states $\vec{\mathbf{x}}_i$, optimizer
$\qquad$ **ParamOpt**$(\hat{\mathcal{L}}, \theta)$.

**for** $i$ *in* **range**$(0, M)$ **do**
$\quad$ **for** $j$ *in* **range**$(0, N)$ **do**
$\quad\quad$ Sample $\vec{\mathbf{x}}'_j \sim g_\theta(\vec{\mathbf{x}}'|\vec{\mathbf{x}}_i)$;
$\quad$ **end**
**end**
$\hat{\mathcal{L}} \leftarrow \frac{1}{MN} \sum_{i,j} \mathcal{L}[\pi; g_\theta](\vec{\mathbf{x}}_i, \vec{\mathbf{x}}'_j)$;
$\theta \leftarrow$ **ParamOpt**$(\hat{\mathcal{L}}, \theta)$;
**for** $i$ *in* **range**$(0, M)$ **do**
$\quad$ Sample $\vec{\mathbf{x}}' \sim g_\theta(\vec{\mathbf{x}}'|\vec{\mathbf{x}}_i)$;
$\quad$ $\alpha \leftarrow \min\{1, \frac{\pi(\vec{\mathbf{x}}')g_\theta(\vec{\mathbf{x}}_i|\vec{\mathbf{x}}')}{\pi(\vec{\mathbf{x}}_i)g_\theta(\vec{\mathbf{x}}'|\vec{\mathbf{x}}_i)}\}$;
$\quad$ Sample $u \sim Uniform[0, 1]$;
$\quad$ **if** $u < \alpha$*:* **then**
$\quad\quad$ $\vec{\mathbf{x}}_i \leftarrow \vec{\mathbf{x}}'$;
**end**

---

distributions. The use of adaptation procedures like Algorithm 2 would introduce uncertainty whether such performance differences were the result of the objective functions themselves and raises the additional question of whether these adaptive procedures are compatible, without modification, with neural MCMC proposal model classes like Equation (3).

It is important to note that adaptation procedures like Algorithm 2 have been developed and validated in the context of relatively restricted proposal model classes. Our results have shown that even objective functions for MCMC efficiency, when developed under assumptions of proposal model class restrictions, can be fundamentally incompatible for optimizing proposal distributions with highly expressive model classes. The problem of model class compatibility also applies to the development of online adaptation procedures. It is still an open question for future research whether these adaptive procedures, originally designed for use with relatively restricted model class proposals, can be applied without modification to the optimization of proposal model classes like Equation (3) or whether some modification is required to maintain compatibility with highly expressive proposals.

### M.1 REPLICATION OF SECTION 7.2 USING ONLINE ADAPTATION

Although the applicability of adaptive procedures like Algorithm 2 to highly expressive proposal distributions remains an open question, we may still test whether the example Ab Initio objective function of Equation (2) is compatible for optimization using these adaptive procedures. To this end, we perform a replication of the experiments of Section 7.2 using the adaptive procedure of Algorithm 2. Aside from the use of Algorithm 2 instead of Algorithm 1, all experimental details remain unchanged from those listed in Appendix J, unless otherwise noted. For optimization, 60000 gradient steps are taken in all cases and Algorithm 2 is used, with $M = 16$ for the German Credit dataset and $M = 8$ for the Heart dataset. The initial states of the persistent Markov Chains are drawn uniformly from the HMC chains originally obtained as specified in Appendix J. In practice, these sampled initial states may be easily obtained via bootstrapping (as used in Song et al. (2017)), either from a less optimized proposal distribution of the same model class or from a traditional MCMC sampling methodology like HMC (as used here). Our experimental results are listed in Table 19.

Table 19: Comparison of sampling performance obtained optimizing augmented multi-scheme proposals of Equation (3) using online adaptive procedure of Algorithm 2 and various objective functions targeting posterior distributions of parameters for logistic regression of UCI datasets. Value means are reported to at most the first significant digit of standard error (reported in parentheses).

| | German Credit (d=21) | | | | Heart Disease (d=14) | | | |
|---|---|---|---|---|---|---|---|---|
| | Acc. Rate | MSJD | ESS per Proposal | | Acc. Rate | MSJD | ESS per Proposal | |
| | | | $x_i$ | $x_i^2$ | | | $x_i$ | $x_i^2$ |
| Ab Initio, Eq. (2) | 0.85(1) | 0.305(4) | 0.61(4) | 0.54(7) | 0.87(1) | 1.05(2) | 0.65(4) | 0.61(5) |
| MSJD Opt. | 0.82(1) | 0.36(1) | 0.64(4) | 0.5(1) | 0.79(1) | 1.38(5) | 0.66(3) | 0.29(5) |
| L2HMC Obj. | 0.84(2) | 0.35(1) | 0.63(4) | 0.54(7) | 0.84(1) | 1.41(3) | 0.76(4) | 0.36(4) |
| GSM-60 | 0.58(3) | 0.23(1) | 0.37(3) | 0.36(2) | 0.60(2) | 0.80(2) | 0.40(3) | 0.42(3) |
| I.I.D. Resample | 1.00 | 0.3700(3) | 0.98(1) | 0.983(3) | 1.00 | 1.222(1) | 0.97(1) | 0.97(2) |

These results using online adaptation are consistent with those obtained using Algorithm 1 previously listed in Section 7.2. In this experiment, the example Ab Initio objective function of Equation (2) is compatible for optimization using the online adaptive update of Algorithm 2. We must still emphasize that future work is required to verify whether online adaptive procedures like Algorithm 2 are generally compatible with highly expressive proposal model classes.

With this in mind, we have two conclusions regarding our results and the employment of adaptive procedures to the optimization of neural MCMC proposals:

- By using Algorithm 1 in lieu of an adaptation procedure and by investigating fundamental properties of MCMC objective functions, our results regarding the compatibility of MCMC objective functions with highly expressive proposal model classes will remain relevant to

future research regarding neural MCMC proposals, regardless of the particular adaptation procedures used in practice or future advancements of the technique of online adaptation.

- The example Ab Initio objective function of Equation (2) is compatible with online adaptive procedures like Algorithm 2, at least for the tasks considered.

- Future research of adaptation procedures like Algorithm 2 is needed to validate their applicability to the optimization of highly expressive proposal distributions. Until such validation is performed, it is difficult to interpret the results of experiments using these adaptive procedures (unmodified from their use with very restricted proposal model classes) for the purpose of comparing the performance of MCMC objective functions and expressive proposal architectures.

