# OpenReview forum: "Semi-Empirical Objective Functions for Neural MCMC Proposal Optimization"
_ICLR.cc/2022/Conference — ICLR 2022 Submitted_

### Official Review · Reviewer_cPU7 · 2021-10-28

**Correctness:** 3
**Technical Novelty And Significance:** 2
**Empirical Novelty And Significance:** 2
**Recommendation:** 3
**Confidence:** 4

**Main Review:**

Finding an effective objective function for training neural MCMC proposals is of great importance for MCMC methods. However, the effectiveness of the proposed Ab Initio objective requires further justifications. Below are my major concerns.

1. Although the motivation for the new objective is quite general, the author ends up with a rather heuristic formulation of the Ab Initio objective matching the optimal acceptance rate of RWM for multivariate Gaussian targets. The general applicability hence is quite doubtful given the relatively simple target distributions used in the demonstrations. The author may try out more complicated models. Also, no significant advantages over MSJD and L2HMC had been found in the experiments.

2. The computation of the Ab Initio objective is not clearly stated. Given that \pi is unknown, the expectation in the first term of L[g;\pi] seems to be intractable in practice.

3. The author claimed (3) can approximate HMC proposals. This should be verified by comparing with the hand-tuned HMC methods.

**Summary Of The Paper:**

This paper proposed an objective function for training neural MCMC proposal, called Ab Initio objective. The new objective seems to improve upon existing alternatives by maintaining property and representation invariance, and reducing the requirement of prior knowledges (e.g., optimal acceptance rate). The author demonstrated the effectiveness of Ab Initio objective on several benchmark models.

**Summary Of The Review:**

The proposed Ab Initio objective seems to be an interesting addition to the current candidates of training objectives for MCMC proposals. However, given the heuristic formulation and inadequate empirical evidence, I am cautious about its general applicability.

---

> ### Author Response · Authors · 2021-11-10
> **Author Response to Reviewer cPU7 (Part 1)**
>
> Thank you for your time and consideration.
>
> As a clarification, as we state in Section 3, "The focus of this work is to illustrate the construction of objective functions such that, when the
> model class G is very expansive (e.g. having been parameterized by a deep generative model), the optimized proposal $g_{opt}$ aligns with our notion of an efficient MCMC proposal" (p. 3).  The purpose of our experiments in Sections 7.1 and 7.2 is to compare the global optima of the considered objective functions when optimizing MCMC proposals parameterized by highly expressive generative models. Our findings show that if someone were to develop a neural MCMC proposal architecture with a relatively unrestricted model class, they could optimize the architecture using the Ab Initio with a degree of confidence not afforded by existing alternative objective functions.  The GSM requires prior knowledge of acceptance rates and the optima of MSJD and L2HMC's objective can be highly inefficient when optimizing highly expressive neural proposals.  This is because the existing objective functions inherently rely on restrictions to the proposal's model class.
>
> ### No significant advantage over MSJD or L2HMC's objective function is found in the experiments:
>
> This is incorrect.  When optimizing the normalizing flow based proposal distribution of Equation (3) in Section 7.1, we find that both MSJD and L2HMC's objective have extremely non-ergodic global minima.  Numerically, this is evidenced in Table 2 by both MSJD and L2HMC's objective attaining 4 orders of magnitude lower ESS relating to uncertainty of the second moments of the marginals of the target distribution.  This is visually explained in Figures 1 (a) and 2, where proposals optimized with MSJD or L2HMC's objective alternate either between the two vertical parts of the target distribution or the two horizontal parts of the target distribution, with an exceedingly low probability of switching between alternating the horizontal and the vertical.  If one were to begin a Markov Chain in a vertical component of the target and attempt MCMC using a MSJD or L2HMC obj. optimized proposal, it would take an excessively large number of samples to reach the horizontal components.   This is highly non-ergodic behavior that is not aligned with our notion of MCMC efficiency.  The Ab Initio objective function does not exhibit this problem.
>
> These same problems with MSJD and L2HMC are again demonstrated with a simpler proposal distribution in Appendix H.  We also find similar (though not as extreme) numerical mismatches between ESS for first and second moments throughout our experiments in Section 7.2 and Appendix J).
>
> Even when optimizing non-neural proposals, we find a reason to prefer using the Ab Initio objective over MSJD.  In Section 7.3, we optimize a preconditioned MALA proposals for the MNIST digit logistic regression target and find that using the Ab Initio objective of Equation 2 yields an optimized proposal with 4 orders of magnitude higher minimal ESS than using MSJD.
>
> Our experimental evidence demonstrates significant advantages of the Ab Initio objective over MSJD and L2HMC's objective.  We argue that these advantages are extremely relevant to the further development of neural MCMC proposals with very expressive model classes.

---

> ### Author Response · Authors · 2021-11-10
> **Author Response to Reviewer cPU7 (Part 2)**
>
> ### The target distributions used in the experiments are quite simple:
>
> The focus of the work is on whether the global optima of MCMC objective functions align with our notion of MCMC efficiency when optimizing neural proposals with very expressive model classes.  Effective experiments for this question need to:
>
>  * Isolate the effect of objective function choice on optimized MCMC performance.
>
>  *  Ensure that the optimized proposals are nearly optimal for the considered objective functions.
>
>
> Our choice of target distributions helps to ensure these goals.  Our use of relatively simple target distributions does not take anything away from our conclusions.  When MSJD and L2HMC's objective optimize a highly expressive neural proposal to yield highly non-ergodic Markov Chains on even the simplest target distribution, this means that MSJD and L2HMC's objective are not aligned with our notion of MCMC efficiency within the model classes of highly expressive neural MCMC proposals.
>
> ### The formulation of the Ab Initio objective is rather heuristic:
>
> We would like some clarification regarding this.  We provide the full theoretical justification for the functional form of the example Ab Initio objective function through Sections 4, 5, and Appendix B.
>
> ### The computation of the Ab Initio objective appears intractable:
>
> We list all of the compared objectives in Appendix F, all include an outer expectation with respect to the target distribution $\pi(x)$.  In practice, this outer expectation is approximated using online adaptation procedures using samples from Markov Chains obtained by the currently optimized proposal distribution.
>
> ### The author claimed Equation (3) can approximate HMC proposals:
>
> We don't just claim it, we show how augmentation with auxiliary variables enables the model class of Equation (3) to approximate HMC in Appendix E (p. 18).  We also show how Equation (3) is capable of approximating many other traditional MCMC schemes.  These are asymptotic properties of the model class of Equation (3).  As part of our discussion regarding Equation (3)'s approximation of HMC, we explicitly state "that we are not claiming that this approximation should be expected to be accurate, merely that it is mathematically possible within the flow proposal’s model class. " (p. 18).
>
> This work is not advocating for the adoption of Equation (3).  The focus of the work is on objective functions.  We are using Equation (3) as an example because it has a very expressive model class that existing objective functions interact poorly with.  Testing and benchmarking the capabilities of the model class of Equation (3) is not the focus of this work.

---

> ### Author Response · Authors · 2021-11-29
> **Additional Author Response**
>
> We responded a few weeks back with clarifications and a few questions of our own but have not received a response.  We still do look forward to discussing and clarifying this work.

---

### Official Review · Reviewer_Etgp · 2021-11-01

**Correctness:** 4
**Technical Novelty And Significance:** 3
**Empirical Novelty And Significance:** 2
**Recommendation:** 5
**Confidence:** 3

**Main Review:**

Strengths:
- The paper tackles the important problem of developing objectives to automatically tune MCMC proposals. This is necessary to tune complex and flexible proposals parameterized by neural networks.
- The first principles provide a sensible guide to design of objectives to tune MCMC proposals. So far different objectives have been proposed without a clear underlying framework. This paper takes a first step in that direction.

Weaknesses:
- While the formulation with the guiding principles is quite general, the only concrete objective proposed is a modification of GSM. Also, even in this case, some parameters are tuned using some ad-hoc methods. For instance, the value of the constant A is set by targeting some optimally known acceptance ratio for a specific setting, and a factor ``d'' is added to correct for the magnitude miss-match between different terms in the objective. Thus, despite the generic presentation, it is not clear to me whether this framework actually simplifies the task of creating new objectives to tune MCMC proposals.
- I'm not fully convinced by the empirical evaluation. The models considered, while illustrative, are quite simple (e.g. Gaussians, low dimensional logistic regression). In addition, the method is not tested for online tuning of MCMC kernel parameters, which is how it would be used in practice. All the evaluations are using true samples from the target, or samples obtained running a long HMC chain. While the authors add this as future work, I think including it in this work would be quite good.

**Summary Of The Paper:**

The paper proposes four desirable properties that should be satisfied by objective functions used to tune MCMC proposals. These are based on first principles, such as that the objective must have a unique global minimum when the proposal is equal to the target, must be representation invariant, and must recover known optimal results in well-studied settings (e.g. optimal acceptance rate for Langevin dynamics with a Gaussian target). Using these principles it proposes a new objective based on GSM (Titsias and Dellaportas, 2019) that satisfies all proposed principles.

**Summary Of The Review:**

Overall I think the whole idea of providing a set of guiding principles for the design of objectives to tune MCMC proposals is interesting and relevant. However, I feel the presentation in this paper may be a bit too generic, and the derivation of the one concrete objective requires a decent amount of manual tuning. Thus, it is not extremely clear to me whether this framework, as presented, simplifies the development of new objectives significantly or not. In addition, I think the concrete objective proposed could be tested more thoroughly, specifically in some settings where samples from the target are not available (i.e. tuning its parameters online as most adaptive MCMC methods do).

---

> ### Author Response · Authors · 2021-11-14
> **Author Response to Reviewer Etgp (Part 1)**
>
> Thank you for your time and consideration.
>
> We will address individual questions below.
>
> ### It is not clear whether this framework simplifies the task of creating objective functions for MCMC proposal optimization:
>
> As we discuss in the introduction, we are motivated to develop objective functions that are suitable for the optimization of highly expressive neural MCMC proposals.  The focus of the work is not to simplify the construction of these objective functions for the sake of simplification.  The focus of the work is to produce objective functions that are suitable for the optimization of neural MCMC proposals with very expressive model classes.
>
> The basic expectation one has for an objective function is that it can yield reasonably correct results (in this case, having optima aligned with our notion of MCMC efficiency) without requiring significant prior knowledge regarding properties of the final optimized result.  Our experiments demonstrate that existing alternative objective functions do not meet these expectations when using a very expressive model class like Equation (3), while the example Ab Initio objective function meets these expectations in the experiments considered. Given the problems demonstrated with existing objective functions on even simple target distributions, we argue that the Ab Initio approximation framework described here is necessary for the proper optimization of highly expressive neural proposals, which itself is necessary for further research regarding the application of highly expressive neural proposals to real world MCMC tasks.
>
> ### The fitting of the coefficient A seems ad-hoc:
>
> The coefficient $A$ is optimized in order to recover analytically known optimal properties in a reference problem (RWM a multivariate gaussian target with an isotropic gaussian proposal in 1,000 dimensions).  Given the nature of this optimization problem (single input and single output with a monotonic relationship between the two in the reference problem of RWM with an isotropic proposal), we found it suitable to perform this optimization manually.  For fitting an Ab Initio objective with more coefficients or using more reference problems, we would advocate a more general (and non-manual) optimization technique, like stochastic Levenberg-Marquardt, as mentioned in the work.
>
> The philosophy we take in this work is that Ab Initio objective functions can serve as functional approximations of the notion of MCMC efficiency in the same manner that neural networks are employed as functional approximations in other problem domains.  As an analogy to the traditional use of neural networks for functional approximation:
>
> * The functional form of an Ab Initio objective function is like the particular neural architecture chosen to perform an approximation.
>
> * The reference problems with analytically known optimal properties used to fit the coefficients in an Ab Initio objective serve as a training set.
>
> * The remaining reference problems with analytically known optimal properties used to verify the Ab Initio objective serve as a test set.
>
>
> With this in mind, we argue that the fitting of the coefficients of an Ab Initio objective are no more ad-hoc than the optimization of the parameters of a neural network in other problem domains.
>
> ### The factor of $d$ seems ad-hoc:
>
> We agree that the choice of $d$ is somewhat arbitrary, to the extent that we could have used $d + 0.001$, $d^{1.001}$, and so on.  However, returning to the neural network analogy, our choice of $d$ is really no different than choosing a particular activation function to use in a neural architecture.  This choice is empirically validated by the dimensional robustness exhibited by the Ab Initio objective in Table 1.  For problems of much larger dimensions, we also explain in Appendix B how the choice of $d$ may be further refined in future work using asymptotic analysis regarding the behavior of the components of the Ab Initio objective function when optimizing for the chosen reference problem with known optimal properties. We argue that the choice of $d$ is no more ad-hoc than much of the design of neural architectures in other problem domains.

---

> ### Author Response · Authors · 2021-11-14
> **Author Response to Reviewer Etgp (Part 2)**
>
> ### The target distributions considered are quite simple:
>
> The considered target distributions are simple.  This allows us to better isolate performance differences between the considered objective functions as being a result of the objective functions themselves and their interactions with the model classes of the proposal distributions.  Our experiments clearly show that, even for these simple target distributions, current objective functions for MCMC proposal optimization cannot suitably optimize highly expressive neural proposals, while the proposed Ab Initio objective can.  From the perspective of future research regarding neural MCMC proposal architectures, the failures exhibited by current alternative objective functions in our experiments are quite fundamental and are clearly linked to detrimental assumptions regarding restrictions to the model class of the optimized proposal. By avoiding these problems, the demonstrated Ab Initio objective allows the optimization of highly expressive neural architectures like Equation (3) with far more confidence than is afforded by existing alternatives.
>
> ### Experiments involving online adaptation should be considered:
>
> We disagree for two reasons.
>
> Firstly, online adaptation serves as an approximation for the outer expectation of the considered objective functions with respect to the target distribution.  Whether or not the optima of these objective functions are aligned with our notion of MCMC efficiency (which is explicitly the focus of our work) is independent of the adaptation procedures used in practice.  Our findings remain relevant to the future development of highly expressive neural MCMC architectures, regardless of the adaptation procedures used for their optimization.
>
> Secondly, our current online adaptation procedures have been developed in a setting with relatively restricted model class proposals.  This work demonstrates that objective functions for MCMC efficiency relying on model class restrictions are unsuitable for the optimization of highly expressive neural proposals.  Whether our current adaptation procedures need to be modified to remain suitable for use with highly expressive neural architectures is an open question.  We suspect that only minor modifications will be needed, but future research into the interaction between adaptation procedures and proposal model class is needed before experiments using online adaptation with highly expressive neural architectures carry much evidential weight.  Carrying out necessary validation of existing adaptation procedures as applied to neural MCMC architectures is beyond the scope of this work.

---

> > ### Comment · Reviewer_Etgp · 2021-11-18
> > **Thanks for the answers**
> >
> > Thanks for answering my concerns. The reply does clarify some things from my original review, in particular the procedure used to fit the parameters of the loss.
> >
> > I still stand by other parts of my original assessment regarding the empirical evaluation. The experiments with simple targets are illustrative, and do reflect weaknesses of existing methods, which the current approach overcomes (at least in the simple cases considered, and using samples from the exact target). However, I think that testing the method on additional, somewhat more complex targets, may be informative too. And I tend to agree with reviewer pwUX about the fact that experiments in an online setting would strongly benefit the paper. These two things may shed light on the practical utility of the method.

---

> > > ### Author Response · Authors · 2021-11-20
> > > **Author Response**
> > >
> > > We have added an additional Appendix (Appendix M) that contains an additional experiment replicating the experiment of Section 7.2 using online adaptation demonstrating that the example Ab Initio objective function of Equation (2) can be compatible with online adaptive procedures.
> > >
> > > ### On the Simplicity of the Target Distributions
> > >
> > > We should point out here that, aside from using a more complex mixture distribution instead of a 2-dimensional multivariate gaussian and excluding Neal's gaussian target (for which i.i.d. resampling lies trivially within the model class of our implementation of Equation (3)), our test target distributions are those used by Titsias and Dellaportas (2019) to validate the GSM.
> > >
> > > ### Additional Experiment with Online Adaptation
> > >
> > > We have performed an additional experiment that replicates the experiment of Section 7.2 using online adaptation.  This online adaptation experiment is included within the newly added Appendix M.  It demonstrates that the Ab Initio objective can be compatible with existing online adaptation procedures.  However, we still emphasize that these adaptation procedures, which were developed for very restricted model classes, do require future research and validation with regards to their fundamental compatibility with highly expressive neural MCMC proposal distributions.

---

### Official Review · Reviewer_pwUX · 2021-11-02

**Correctness:** 4
**Technical Novelty And Significance:** 3
**Empirical Novelty And Significance:** 2
**Recommendation:** 8
**Confidence:** 3

**Main Review:**

# Strengths

I think the problem was amazingly motivated. The description of the objective function was very clear: both in terms of its description as well as its need. I am also fascinated that for a fixed $A$, the proposed approach is able to reproduce classic theory regarding the optimal acceptance probability for random walk metropolis and metropolis adjust langevin diffusion.

# Weaknesses

I first want to note that the length of the weakness section relative to the strength section does not reflect my opinion of the paper. While I'm a big fan of the approach I'm somewhat confused about the actual proposed method. First, the proposed objective function is an expectation with respect to the target density, $\pi(x)$. While I understand that theoretically, this is important, practically this is *very* restrictive as one either needs to have access to the target density--which if this is the case, why do MCMC--or one needs to run access to approximate samples, but if you have access to these samples why run this method.

The second thing that isn't clear is the actual implementation: is the proposal optimized before running the MCMC sampler or is it optimized while the MCMC algorithm is running? Also, in Appendix D it says "For optimization, 20000 gradient steps are taken, with each training batch taken from a single starting point sampled i.i.d. from the target distribution from which 50 proposals are generated to calculate expected loss". I'm confused about what is meant by 50 proposals? Is the gradient estimated using 50 samples? Is the sampler run for 50 time steps, and those samples are used to approximate expectations? This same question is also relevant to the experiment section as well. The confusion around proposals makes it much harder to understand what is actually going on.

Lastly, I have some comments about the experiments. I think for all quantities, it would be useful to state whether higher/lower is better.
In experiment 7.1, the authors state "Position dependence within the proposal distribution lead MSJD and L2HMC’s modification to optimize towards arbitrarily non-ergodic proposal distributions, sampling only either the horizontal or vertical components of the target distribution.". From this description, I would expect the MJSD and L2HMC to not mix well but according to the results in Table 3, they both have very high ESS (also I thought ESS was upper bounded by 1?) and MSJD. I noticed a similar discrepancy in 7.3 where the authors state "The resampling normalizing flow did not produce a single accepted proposal within these evaluation Markov Chains, hence we are unable to estimate ESS and our estimates of acceptance rates and MSJD for this proposal distribution should be viewed as highly uncertain." but the resampling normalizing flow has the highest Ab Initio! Also, I'm confused about what resampling normalizing flow is (I also don't think it was described anywhere in the paper).

# Comments
1. There should be a section on how the ESS was computed.
2. It would be beneficial to the reader if MSJD was defined.

**Summary Of The Paper:**

The authors design an objective function from first principles that can be used to optimize proposal distributions for MCMC. They empirically verify that their objective function with a fixed hyperparameter--tuned a priori on another task--can reproduce theoretical results over a variety of target distributions.

**Summary Of The Review:**

While I like the motivation of the paper, I think the actual algorithm and the experimental details are somewhat opaque. This can be easily fixed by fixing the manuscript and making these things very clear. I'm also somewhat concerned about the practicality of the algorithm and I think this needs to be addressed by the authors in the manuscript. For these reasons, I lean towards a weak reject though I am more than happy to increase my scores if the authors address my concerns.

---

> ### Author Response · Authors · 2021-11-14
> **Author Response to Reviewer pwUX (Part 1)**
>
> Thank you for your time and consideration.
>
> We will address individual questions below.
>
> ### How is the expectation with respect to $\pi(x)$ handled in practice?
>
> We'll note that the outer expectation with respect to the target distribution, $\pi(x)$, appears in all of the compared objective functions (see our listings of the functions in Appendix F) and is not unique to the Ab Initio objective function demonstrated in this work.  In practice, this outer expectation is approximated over the course of optimizing the objective functions using online adaptation procedures.  These adaptation procedures work by specifying some number of persistent Markov Chains that are used to approximate samples from $\pi(x)$ during training.  A general summary of these adaptive procedures is to approximate your samples from $\pi(x)$ based on the current states of the persistent Markov Chains, draw samples from your proposal, $g_{\theta}(x'|x)$, and perform a gradient step, and then advance the state of the Markov Chains using a Metropolis-Hastings acceptance step (either from the samples of $x'$ just drawn or from samples drawn based on the newly updated proposal).
>
> These adaptation procedures were developed for use in settings with relatively restricted model classes.  This work has shown how even objective functions for MCMC efficiency can interact detrimentally with highly expressive neural proposals when originally designed for use with restricted model class proposals.  Future research is needed to verify whether our current adaptation procedures remain applicable for use with very expressive proposal model classes or whether some modification to the procedures is needed.  We argue that analyzing and validating adaptation procedures lies beyond the scope of this work, as our findings regarding the interaction between MCMC objective functions and proposal model classes remain relevant regardless of the particular adaptation procedure used in practice.
>
> It was for these reasons that we left the topic of adaptation procedures for future research within the conclusion.
>
> ### Clarifications regarding the optimization and evaluation of compared objective functions:
>
> For our experiments, the proposal distributions are optimized before running the MCMC sampler to evaluate their performance.  To help clarify this, we have changed "For each optimization..." to "After each optimization..." in sections 7.1 and 7.2.  We have changed "For each evaluated scheme, we determine..." to "For each evaluated scheme, after optimization using the respective objective function, we determine..." in Section 7.3.  We have added similar clarifications to Appendices G, I, J, and K.
>
> When forming empirical expectations to estimate the compared objective functions during optimization, we use the following process at each gradient step: draw M samples either directly from $\pi(x)$ or from the long equilibrated HMC chains and then draw N samples from $g_{\theta}(x'|x)$ starting from each of the M samples approximately drawn from $\pi(x)$ to form an empirical estimate containing a total of MN samples. To help clarify this, the relevant sentence in Appendix D now reads "For optimization, 20000 gradient steps are taken, with each training batch taken from a single starting point sampled i.i.d. from the target distribution from which 50 independent proposals from $g_{\theta}(x'|x)$ are generated to empirically estimate expected loss on the basis of 50 total samples.".  Appendices G, I, J and K have been similarly clarified.  For example, the related sentence in Appendix G now reads "For optimization, 20000 gradient steps are taken, with each training batch taken from 8 starting points sampled i.i.d. from the target distribution from each of which 100 independent proposals from $g_{\theta}(x'|x)$ are generated to empirically estimate expected loss on the basis of 800 total samples.".
>
> Thank you very much for pointing this out, we do genuinely appreciate this feedback.  Do you believe these changes would have prevented the original confusion?

---

> > ### Comment · Reviewer_pwUX · 2021-11-16
> > **Response to part 1**
> >
> > Thank you so much for getting back to me. I am on the same page that while this work can be extended to the online adaptation of MCMC samplers, this work focuses on training the proposal before running the MCMC sampler.
> >
> > I think I am somewhat still a little confused about the actual training procedure.
> > 1. In the loss function one needs to compute $\mathbb{E}_{\pi(x)} [KL( g(x' \vert x) \Vert \pi(x') ) ] $. From the response, it sounds like to approximate the expectation wrt $\pi(x)$, you sample directly from $\pi(x)$ or from an HMC chain. Is this correct?
> > 2. I'm confused about the way the word proposal is being used. For instance, in the response, it was said "...from which 50 independent proposals from $g_\theta(x' \vert x)$  are generated...". Do you mean that the chain was run for 50-time steps? If that's the case then these samples can't be independent of one another.

---

> > > ### Author Response · Authors · 2021-11-16
> > > **Author Response**
> > >
> > > Thank you for the questions.
> > >
> > > All of the considered objective functions can be expressed as having an outer expectation with respect to $\pi(x)$ and an inner expectation with respect to $g(x'|x)$ , having the general form $E_{x \sim \pi(x)} [ E_{x' \sim g(x'|x)} [ ...]]$.  With this in mind:
> > >
> > > 1.  Yes, that is correct.  Strictly for the purposes of our experiments, we approximate the outer expectation by samples from $\pi(x)$ or by samples from an HMC chain (when direct sampling was unavailable).
> > >
> > > 2. We are approximating the inner expectations using some number of independent samples from $g(x'|x)$ for each $x \sim \pi(x)$ sampled previously. So when we say "from which 50 independent proposals from  $g_{\theta}(x'|x)$ are generated", we mean we are drawing 50 samples of $x'$ from $g_{\theta}(x'|x)$ for each $x$ having been previously sampled.  We do not mean that we are running a full Markov Chain with Metropolis-Hastings corrections for 50 time steps.  Do you believe adding "(for the purpose of empirically estimating inner expectations with respect to $g_{\theta}(x'|x)$ within the compared objective functions)" to these descriptions will avoid similar confusion?

---

> > > > ### Comment · Reviewer_pwUX · 2021-11-16
> > > > **Response**
> > > >
> > > > Aha, okay now I'm on the same page!
> > > >
> > > > I think what would really make it clear is having an algorithm table!
> > > >
> > > > Okay, now that I'm on the same page I want to go back to one of my original points in my initial review. I was initially concerned with practicality and now that my understanding has been clarified I am still concerned but I think my initial concern was based on me not reading it from a theoretical point of view. I think this may also be due to some of the references where they focus on online adaptation of the proposal i.e., Titsias and Dellaportas, NeurIPS 2019. I think this makes this point clear, this should be discussed in the experiments section where you reiterate that to disentangle the benefits of the proposed function from the nuances of online adaptation, you focus on cases where you have access to $\pi(x)$. I do think though the paper would benefit highly from a toy example where adaptation is done online!

---

> > > > > ### Author Response · Authors · 2021-11-20
> > > > > **Author Response**
> > > > >
> > > > > We have added an additional Appendix (Appendix M) that contains an algorithmic specification (Algorithm 1) of how we perform optimization throughout our experiments, the explanation for why we use this over online adaptation for the purposes of our experiments, and an additional experiment replicating the experiment of Section 7.2 using online adaptation demonstrating that the example Ab Initio objective function of Equation (2) can be compatible with online adaptive procedures.
> > > > >
> > > > > ### Textual Clarifications
> > > > >
> > > > > We have added a reference to this new appendix within the general explanation of our experimental approach at the beginning of Section 7 on page 6, reading "A comparison of our optimization procedure with traditional online adaptive techniques is provided in Appendix M.".
> > > > >
> > > > > Within Appendices D,G,I, and J, we have added references to the algorithmic listing of our optimization procedure to further clarify our approach.  For example, the relevant section of Appendix G now reads: "... taken from 8 starting points sampled randomly from HMC Markov Chain and from each starting point 100 independent proposals
> > > > > from $g_{\theta}(x' |x)$ are generated to empirically estimate expected loss on the basis of 800 total samples. The gradient update steps take the form of Algorithm 1 with M = 8 and N = 100."
> > > > >
> > > > > ### Additional Experiment with Online Adaptation
> > > > >
> > > > > We have performed an additional experiment that replicates the experiment of Section 7.2 using online adaptation.  This online adaptation experiment is included within the newly added Appendix M.  It demonstrates that the Ab Initio objective can be compatible with existing online adaptation procedures.  However, we still emphasize that these adaptation procedures, which were developed for very restricted model classes, do require future research and validation with regards to their fundamental compatibility with highly expressive neural MCMC proposal distributions.

---

> > > > > > ### Comment · Reviewer_pwUX · 2021-11-29
> > > > > > **Response**
> > > > > >
> > > > > > Great, thank you so much! I think this work is a valuable contribution and will increase my score appropriately.

---

> > > > > > > ### Author Response · Authors · 2021-11-29
> > > > > > > **Author Reply**
> > > > > > >
> > > > > > > Thank you very much for your helpful feedback.  We look forward to the update.

---

> ### Author Response · Authors · 2021-11-14
> **Author Response to Reviewer pwUX (Part 2)**
>
> ### It would be good to include a higher/lower is better for the compared quantities:
>
> We have amended the caption of Table 6 to now read "Comparison of sampling performances and Ab Initio losses (lower is better)...".  A lower Ab Initio loss is better, with the lowest possible value being 0 as obtained by exact I.I.D. resampling from the target distribution.  For Section 7.3, the resampling normalizing flow is the worst performing with respect to Ab Initio loss while the preconditioned MALA proposal optimized using Equation (2) is the best performing with respect to Ab Initio loss.
>
> For the other statistical quantities, like MSJD and ESS, higher or lower values are not necessarily better.  For these quantities, values closer to those obtained by exact I.I.D. resampling are better, which is the reason we included the values from (approximate) I.I.D. resampling within these tables for comparison.
>
> To note, ESS is not bounded from above by 1.  For single chain estimates, the ESS per proposal may be estimated via:
>
> $$ESS \  per \  proposal= \frac{1}{1 + 2\sum_{t=1}^{\infty}\rho_{t}}$$
>
> Where $\rho_{t}$ is the autocorrelation at lag $t$.  For many typical MCMC schemes, this autocorrelation remains positive and the ESS per proposal is typically found below 1.  However, when this autocorrelation is negative, the estimated ESS per proposal can rise above 1.  Because negative autocorrelation is a departure from exact i.i.d. resampling from the target (where there is no autocorrelation), we cannot make the general statement that a higher or lower ESS is always better.
>
> ### Why is ESS relating to uncertainty of estimating the first moments of the target distribution so high for MSJD and L2HMC Obj. in Section 7.1?
>
> The above mentioned negative autocorrelation occurs with the MSJD and L2HMC obj.  optimized proposals in Section 7.1. These objectives optimize towards proposals that nearly exclusively alternate left-right or up-down within the target mixture distribution, leading to negative autocorrelation when estimating the first moments of the marginals of the target.   This is why their ESS per proposal results relating to estimating first moments (listed under $x_{i}$) rise above 1 in Table 6.
>
> The non-ergodicity of the MSJD and L2HMC obj. optimized proposals in Section 7.1 is shown numerically by their ESS per proposal results relating to estimating second moments (listed under $x_{i}^{2}$)  in Table 6, which are orders of magnitude lower. Because these proposals alternate left-right or up-down with very little chance of switching over, half of the probability mass of the target remains effectively unexplored by Markov Chains following from these proposals.  Using these proposals, exceedingly long equilibration times are needed to accurately estimate expectations of the form $\mathbb{E}[x_{i}^{2}]$, which is quantified by the second moment related ESS measurements.
>
> ### The resampling normalizing flow in Section 7.3 is not defined:
>
> The resampling normalizing flow is an unconditional normalizing flow that is optimized to directly approximate $\pi(x)$.  It's first mention in Section 7.3 now reads "The resampling normalizing flow (an unconditional normalizing flow optimized to directly approximate the target distribution)...".
>
> ### A section on the calculation of ESS would be helpful:
>
> We have added a new appendix (Appendix L) summarizing the calculation and interpretation of ESS results.  We have consolidated some of the discussion of ESS from the other appendices into this new appendix.
>
> ### Is MSJD defined within the work?
>
> The acronym was defined at first use in Section 2: "Mean squared jump distance (Pasarica and Gelman, 2010) (MSJD) remains...".  It's numerical definition was also provided within Appendix F.

---

> > ### Comment · Reviewer_pwUX · 2021-11-16
> > **Response to part 2**
> >
> > Thanks so much, this clarified things a lot!

---

### Official Review · Reviewer_qDJ6 · 2021-11-03

**Correctness:** 4
**Technical Novelty And Significance:** 3
**Empirical Novelty And Significance:** 3
**Recommendation:** 5
**Confidence:** 4

**Main Review:**

Although the paper approaches an important problem in the MCMC field, its contribution is incremental. One may consider it as a special case of the objective from [Titsias, 2019]. The proposed approach has several major issues:
- The conjecture of the existence of universal $\beta$ is not well-supported. The authors motivate the choice of GSM objective by verifying several properties. However, further developments are based on the conjecture that there should be some optimal tradeoff between the acceptance rate and the proposal entropy for all target distribution and all proposals. This conjecture is validated neither theoretically nor empirically.
- The way of finding $\beta$ by "manual trial and error" is not acceptable. Although the authors mention that it is possible to estimate the constant using some grounded techniques, they describe their procedure as "trial and error". This fact cannot persuade practitioners to use the proposed objective. The constant should be estimated at least by using grid search demonstrating the performance of the algorithm for different values. Furthermore, this empirical study has to be extended to different proposals and different targets demonstrating the universality of the constant. Finally, the variance of the estimation should be reported.
- The evaluation of the proposed approach is not practical. To be more precise, the proposed objective requires samples from the target distribution. Throughout the experiments, the authors collect these samples separately running a long HMC chain. However, obtaining a large set of uncorrelated samples from the target using HMC could be impossible in practice. Not to mention that this is the final goal of any MCCM algorithm. Note that in [Titsias, 2019], this issue is bypassed by collecting the samples online and treating them as samples from the target distribution. This setting is very important for practical applications and should be explored empirically.

Other comments:
- Section 7.2 is not motivated. Indeed, some MCMC algorithms use augmented space for sampling. However, the reason for using it here is unclear. This section could give more insights to the reader if two algorithms have been compared (with augmented space and without).
- Section 7.3 is unclear. The authors conclude that the proposed objective yields a robust measure of MCMC efficiency and can be used as a metric for comparison. However, this conclusion is not argued. In all previous examples, the authors rely on ESS to measure the efficiency of chains and motivate the effectiveness of their objective also by ESS. Thus, we have a question why would one use the proposed objective instead of ESS to measure the efficiency?


**Summary Of The Paper:**

The paper approaches the problem of learning the optimal parameters of the MCMC chain. Namely, it tries to design an objective function that allows for the learning of a perfect proposal distribution in the Metropolis-Hastings algorithm. The main contribution of the current paper is based on the developments from [Titsias, 2019]. [Titsias, 2019] proposes the following objective for the optimization of the proposal
$$\text{GSM} = \text{Acceptance Rate} + \beta \cdot \text{Entropy of the proposal}.$$
Maximization of this objective results in better mixing properties of the chain. Indeed, the next sample comes from a distribution with high entropy and should be accepted with a high probability, and the proposal should converge to the independent sampler in the perfect scenario.

However, the choice of $\beta$ remains to be a question since it defines the tradeoff between the acceptance rate and the entropy, which is known to be different for different proposals and different targets. The original paper [Titsias, 2019] solves this issue by adaptively setting the acceptance rate to some desired value.

The current paper then can be summarized as follows. Instead of setting $\beta$ adaptively based on the acceptance rate, the authors propose to set it constant. The constant is then found by "manual trial and error" in such a way that the acceptance rate of Random-Walk Metropolis-Hastings is 0.234 for high-dimensional standard Gaussian, which is known to be optimal for this task. Then this constant is claimed to be universal for all targets and proposals. To verify this empirically, the authors first optimize the parameters of MALA and RW Metropolis-Hastings for several synthetic targets (Uniform, Laplace, Cauchy, Gaussian) and demonstrate that the procedure yields the right parameters.

Finally, the authors perform an empirical study of the proposed procedure. They use the proposed objective to learn a proposal parameterized by the normalizing flow and compare it to the optimization of the original objective from [Titsias, 2019] targeting different acceptance rates. The target distributions for the empirical study are a mixture of four Gaussians (2-D), Bayesian logistic regression (21-D, 14-D).

[Titsias, 2019]: Titsias, Michalis, and Petros Dellaportas. "Gradient-based adaptive markov chain monte carlo." Advances in Neural Information Processing Systems 32 (2019): 15730-15739.

**Summary Of The Review:**

The paper proposes and analyses a special case of the objective from [Titsias, 2019]. Unfortunately, the empirical study is not sufficient to make positive conclusions about the proposed technique.

---

> ### Author Response · Authors · 2021-11-10
> **Author Response to Reviewer qDJ6 (Part 1)**
>
> Thank you for your time and consideration.
>
> To address a serious misunderstanding of our work, our work is summarized here as attempting to find a universal $\beta$ within the GSM by Titsias and Delaportas.  We do not.
>
> We lay out a series of properties that we argue an objective function of MCMC efficiency should follow (Section 4).  The GSM does not follow these properties, as it is neither proper nor representation invariant.  We therefore build from the GSM by adding a cross entropy term terms to derive an objective function satisfying our proposed properties (Appendix B).  In the introduction, we clearly state that "Our example Ab Initio objective function of Equation (1) takes inspiration from the components of the GSM, while also ensuring the functional limitations argued for in Section 4" (p. 3).  When introducing Equation (1) in Section 5, we explain that "For this work, we adapt the GSM into an Ab Initio objective function (details are provided in Appendix B) to follow the constraints of Section 4" (p. 4).  The example Ab Initio objective function that we use throughout the paper (Equations (1) and (2)) is \textit{not} the GSM, as proposal entropy is replaced by KL divergence, which may be further verified by our listing of all compared objective functions in Appendix F.
>
> Stating that our work proposes and analyses a special case of the GSM is completely incorrect and egregiously misrepresents the content of the work.
>
> ### The conjecture of a universal $\beta$ is not well supported:
>
> We make no conjecture regarding the existence of such a universal coefficient that will remain optimal for all proposal and target distributions.  As we state in the conclusion "we do not presume that the particular example
> Ab Initio objective function of Equation (2) will be absolutely universal" (p. 9).  We clearly state in section 3 that "The focus of this work is to illustrate the construction of objective functions such that, when the
> model class G is very expansive (e.g. having been parameterized by a deep generative model), the optimized proposal $g_{opt}$  aligns with our notion of an efficient MCMC proposal" (p. 3).  To this end, we propose approximating the ground truth notion of MCMC efficiency "$\mathcal{L}^{\*}$ by the combination of simpler objective functions as weighted by hyperparameter coefficients ..." that "can then be fit so as to recover optimal properties established within existing theoretical works" (p. 4).  The approach is similar to a series approximation of $\mathcal{L}^{\*}$.
>
> ### The use of manual trial and error is not acceptable:
>
> To clarify, by "manual trial and error", we meant a manual approximate Newton-Rhapson optimization.  We have clarified our description within the current draft.  We stand by the use of this methodology for optimizing the hyperparameter for Equation (2), as there is a single optimized parameter, a single optimized output, and for isotropic RWM proposals the acceptance rate is monotonically dependent on A in Equation (2).  Regardless of how the hyperparameter coefficients are obtained, the main question is whether they are able to robustly optimize neural MCMC proposals with expressive model classes.  Our experimental results throughout the work serve this purpose.
>
> ### The use of HMC chains is impractical:
>
> We state in Section 3 “the focus of this work is to illustrate the construction of objective functions such that, when the model class G is very expansive (e.g. having been parameterized by a deep generative model) the optimized proposal $g_{opt}$ aligns with our notion of MCMC efficiency” (p. 3).  In Section 7, we clearly explain the use of the HMC chains in our experiments for the focus of our work, stating “As we are comparing the properties of the global optima of these objective functions, we estimate sampling from the target distribution by either direct sampling or by sampling from a long equilibrated HMC chain” (p. 6).  Our experiments in sections 7.1 and 7.2 are evaluating the compared objective functions on the basis of their compatibility for optimizing MCMC proposals with highly expressive model classes.  Our experiments demonstrate that, for highly expressive neural MCMC proposals, existing MCMC objective functions (the GSM, MSJD, L2HMC's objective function) have very significant drawbacks compared to the example Ab Initio objective function.  The GSM requires prior knowledge of optimal acceptance rates.  MSJD and L2HMC's objective have global minima with very poor MCMC performance when optimizing expressive neural proposals. These drawbacks exist independently of adaptation strategies.

---

> > ### Comment · Reviewer_qDJ6 · 2021-11-29
> > **Rebuttal response**
> >
> > Thank you for the clarifications!
> >
> > 1. Indeed, I was wrong in claiming that this is the same objective as proposed in [Titsias, 2019]. In my mind, I was taking the integral $\int dx g(x'|x)\pi(x) = \pi(x')$, which is obviously wrong since $g(x'|x)$ is the proposal. That's why I agree that the authors propose a novel objective based on the mentioned properties. Thank you for the clarification!
> >
> > 2. Regarding the selection of $\beta$. We might disagree on the wording here but the usage of the single $\beta$ value for all proposals and target distributions is a fact. Moreover, the consistency of this value is not verified even empirically. I would be more convinced if the authors obtained the same value on several tasks and proposals (up to the same number of significant figures). Such an experiment would be a great confirmation of the discussed theoretical properties of the objective.
> >
> > 3. I would suggest the authors avoid such wordings as "manual trial and error" and "manual approximate Newton-Raphson optimization".  A clear statement of the used procedures is essential for the reproducibility of the results. I understand that some methods could be performed manually, but they still require a rigorous description (maybe even more rigorous than widely-used implementations).
> >
> > 4. Regarding the usage of samples from the true target distribution. I understand that the experiments illustrate the properties of the proposed objective function and its competitors. However, I think the scenario of the comparison should be practical since the ultimate goal of the paper is to propose a practical objective for learning proposals and chains comparisons.
> >
> > To sum up, I acknowledge that my first assessment of the work was too pessimistic; however, I still find the paper to be below the acceptance threshold. I would like to encourage the authors to strengthen their empirical study and finish their project since I consider it to be important for the field.

---

> > > ### Author Response · Authors · 2021-11-29
> > > **Author Response (Part 1)**
> > >
> > > Thank you for your response, we address individual comments below:
> > >
> > > ### On the Distinction with the GSM
> > >
> > > To be frank, we are confused as to how the integral $\int dx g(x'|x)\pi(x)$ would have lead to this misunderstanding, as it never appears within our work and it is not a component of any of the considered objective functions.
> > >
> > > The definition of the Ab Initio objective function and how it relates to the GSM is discussed throughout the work.
> > >
> > > It seems to us that a glance at the definitions of our Ab Initio functions in Equations (1) and (2), our discussions of how the Ab Initio objective function relates to the GSM within Sections 2, 4, and Appendix B, or to our descriptions of the compared objective functions in Appendix F would have prevented this misunderstanding.
> > >
> > > ### On the Fitting of Coefficients
> > >
> > > The difference between "a set of coefficients is presumed universally optimal" and "a set of coefficients is fit to provide a good approximation" is more than a difference in wording.
> > >
> > > As explained in Section 5, Ab Initio objective functions are approximations of our notion of MCMC efficiency that with coefficients that are fit using reference MCMC problems with analytically known optimal solutions.  We do not rely on some conjecture that such coefficients will be universal. The philosophy we take in this work is that Ab Initio objective functions can serve as functional approximations of the notion of MCMC efficiency in the same manner that neural networks are employed as functional approximations in other problem domains.  The analogy to neural networks is as follows:
> > >
> > >
> > > * The functional form of an Ab Initio objective function is like the particular neural architecture chosen to perform an approximation.
> > >
> > > * The reference problems with analytically known optimal properties used to fit the coefficients in an Ab Initio objective serve as a training set.
> > >
> > > *  The remaining reference problems with analytically known optimal properties used to verify the Ab Initio objective serve as a test set.
> > >
> > >
> > > When training neural networks, we do not assume that the optimized parameters from training data represent universally optimal parameters on test data.  Simply because a neural network is optimized to a single set of optimal parameters on a given training set does not mean that it's use on test data is based on a presumption that those parameters must be universal.  Instead, we find that with proper training the optimized parameters from training data represent suitably good approximations for application on test data.  Therefore, this work is not relying on the presumption of a universal set of coefficients for Ab Initio objective functions that are optimal for all target and proposal distributions any more than our use of neural networks relies on a presumption of a universally optimal set of parameters (which it doesn't).

---

> > > ### Author Response · Authors · 2021-11-29
> > > **Author Response (Part 2)**
> > >
> > > ### On the Reproducibility of Newton-Rhapson Optimization
> > >
> > > As the optimization problem in question has a single input and a single output with monotonic relation between them and the Newton-Rhapson procedure is a standard algorithm in such settings, we do not believe this represents a significant replicability concern.
> > >
> > > ### Comparisons of Objective Functions Should be Performed Under an Adaptive Setting
> > >
> > > We disagree, for two reasons.
> > >
> > > Firstly, online adaptation serves as an approximation for the outer expectation of the considered objective functions with respect to the target distribution.  Whether or not the optima of these objective functions are aligned with our notion of MCMC efficiency (which is explicitly the focus of our work) is independent of the adaptation procedures used in practice.  Our findings remain relevant to the future development of highly expressive neural MCMC architectures, regardless of the adaptation procedures used for their optimization.
> > >
> > > Secondly, our current online adaptation procedures have been developed in a setting with relatively restricted model class proposals.  This work demonstrates that objective functions for MCMC efficiency relying on model class restrictions are unsuitable for the optimization of highly expressive neural proposals.  Whether our current adaptation procedures need to be modified to remain suitable for use with highly expressive neural architectures is an open question.  We suspect that only minor modifications will be needed, but future research into the interaction between adaptation procedures and proposal model class is needed before experiments using online adaptation with highly expressive neural architectures carry much evidential weight.  Carrying out a complete validation of existing adaptation procedures as applied to neural MCMC architectures is beyond the scope of this work.
> > >
> > > Since the initial submission, we have added an additional appendix (Appendix M) that performs a replication of the experiments of Section 7.2 using an online adaptive algorithm with compatible results.  As the current adaptive methods are not verified for application to highly expressive proposals, comparisons such as those provided in Appendix M do not directly relate the capabilities of the objective functions themselves (until verified, existing adaptive procedures must be viewed as potentially confounding factors to such analysis).  However, the results of Appendix M do indicate that the example Ab Initio objective can be compatible with existing online adaptive procedures.

---

> ### Author Response · Authors · 2021-11-10
> **Author Response to Reviewer qDJ6 (Part 2)**
>
> ### Section 7.2 is not well motivated:
> We disagree.  The focus of the work is to develop objective functions that can be confidently used to optimize highly expressive neural MCMC architectures.  Augmentation is a common feature of existing, traditional MCMC schemes and we should assume that the trend will continue, with new augmented neural architectures being developed.  If we are concerned about whether objective functions are compatible with neural MCMC proposal classes, then we should test augmented neural MCMC proposals.
>
> Because the focus of our work is on comparing the differences between objective functions (all else being fixed), we will not compare the performance between different proposal distributions.
>
> ### Section 7.3 is unclear:
>
> Our conclusion that the example Ab Initio objective is generally robust and can serve as a measure of MCMC efficiency is empirically demonstrated by our results in sections 6, 7.2, 7.3, Appendix I, and Appendix J. ESS relating to the first moment provides an indication of how faithfully MCMC estimates will converge towards means, ESS relating to the second moment give indications regarding how faithfully the MCMC estimates will converge towards variances, etc. .  These ESS estimates do not provide a scalar value summarizing sampling efficiency for "arbitrary" use.  Good ESS performance in first moments can mask poor ESS performance in second moments (as exemplified by the behavior of the MSJD and L2HMC objectives in Table 2).  The Ab Initio objective offers a single scalar alternative that, based on our experimental results, aligns well with our notion of MCMC efficiency.  Because the GSM requires a fixed target acceptance rate, it cannot be used to compare different proposal architectures in this manner.

---

### Decision · Program_Chairs · 2022-01-20

**Decision:**

Reject

**Comment:**

This paper proposes guiding principles with which to design objective functions for proposal distributions for MCMC. They design one such objective based on GSM (Titsias and Dellaportas, 2019). The two concerns raised by reviewers that resonated the most with me were:

- it was not clear that the actual proposed objective was the best way to implement these guiding principles
- a weak empirical evaluation that did not consider online tuning and high-dim, highly non-Gaussian targets.

After rebuttal, revision, and discussion, reviewers felt that the authors did a reasonable job of addressing the issue of online tuning, but very highly non-Gaussian targets were not addressed. There was still a sense that the ultimate instantiation of the design principles was a somewhat adhoc loss. Ultimately, I think that this work is just below the bar for acceptance and it can be improved by clarifying the choices made in implementing the objective and some more ambitious experiments.